# Anti-Backdoor Coreset Selection via Cumulative Entropy

Qi Zhao [1]   Christian Wressnegger [1]

## Abstract

Recent training-time defenses against neural back-doors isolate a benign subset from poisoned training data, to learn a backdoor-free model from it. In this paper, we formulate this defense strategy as a coreset selection problem, giving rise to so-called *"Anti-Backdoor Coreset Selection."* Since poisonous samples have a) lower prediction uncertainty and are b) less frequent than benign samples, coreset selection naturally focuses more on samples associated with benign functionality than the backdoor functionality. We use the Cumulative Entropy as selection criterion to further facilitate this effect. The metric tracks the learning dynamics of training samples and allowing us to select benign samples with high informativeness for the coreset. Additionally, we unlearn the chosen samples in each epoch to facilitate the separability between benign and poisonous samples. Together, this yields an exceptionally effective training-time defense that constructs a benign coreset to train a backdoor-free model. Unlike prior defenses that compromise natural accuracy and fail against certain attacks, our method mitigates backdooring attacks consistently with a negligible impact on natural performance. The implementation of our method is publicly available at: https://intellisec.de/research/abcs.

## 1. Introduction

Effectively training deep neural networks requires large amounts of training data (LeCun et al., 2015; Hestness et al., 2017; Sun et al., 2017). However, manually vetting these training datasets is infeasible, so that, in practice, models are frequently learned using data without any security guarantees (Carlini, 2021; Carlini et al., 2024). This circumstance gives rise to data-poisoning attacks (Biggio et al., 2012; Big-gio & Roli, 2018), e.g., to introduce neural backdoors (Gu et al., 2017; Chen et al., 2017; Liu et al., 2018; Nguyen & Tran, 2020; 2021; Turner et al., 2019; Shafahi et al., 2018; Zhao et al., 2023; Barni et al., 2019; Jha et al., 2023).

A neural backdoor establishes a shortcut from a trigger pattern to a target prediction (Biggio & Roli, 2018; Gu et al., 2017; Chen et al., 2017), which can be established in different ways: So-called *dirty-label attacks* manipulate the training samples (to add the trigger pattern) and change their labels (to force the target prediction) (Gu et al., 2017; Chen et al., 2017; Liu et al., 2018; Nguyen & Tran, 2020; 2021). *Clean-label attacks*, in turn, go a step further and do not change the ground-truth label but manipulate samples from the target class to strengthen the connection to the trigger pattern (Turner et al., 2019; Shafahi et al., 2018; Zhao et al., 2023; Barni et al., 2019).

Recently, the community has focused on tackling data poisoning at its root, that is, learning a clean, backdoor-free model despite the training dataset being poisoned (Li et al., 2021b; Zhang et al., 2023; Huang et al., 2022a; Gao et al., 2023; Chen et al., 2022; Zhu et al., 2023b; Zhao & Wressnegger, 2025). This setting poses several challenges beyond mitigating the backdoor: First, the natural performance (the accuracy on the primary, aimed for task) should match that of a model trained on a not poisoned dataset. Meanwhile, performance must not decline even if the training data is not poisoned. Second, the defense should not rely on a clean reference dataset as this is tied to manual vetting effort. Finally, a defense should not significantly increase training time. Prior work unfortunately falls short in at least one of these criteria or even fails to mitigate the backdoor at all.

In this paper, we consider an alternative viewpoint on training-time defenses. Most approaches (Huang et al., 2022a; Gao et al., 2023; Chen et al., 2022) aim for splitting the training dataset in poisoned and benign (not poisoned) data to learn a backdoor-free model on the benign subset. We formulate this strategy as a coreset selection problem (Coleman et al., 2020; Toneva et al., 2019; Hekmat-far & Farahani, 2009; Ducoffe & Precioso, 2018; Sener & Savarese, 2018; Mirzasoleiman et al., 2020; Killamsetty et al., 2021; 2020) and propose a new defense scheme: *"Anti-Backdoor Coreset Selection (ABCS)"*. Our method follows the intuition that the training data encodes two different

[1]KASTEL Security Research Labs, Karlsruhe Institute of Technology (KIT) . Correspondence to: Qi Zhao <qi.zhao@kit.edu>.

*Proceedings of the 43rd International Conference on Machine Learning*, Seoul, South Korea. PMLR 306, 2026. Copyright 2026 by the author(s).

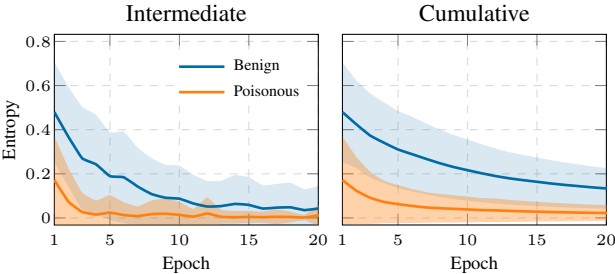

*Figure 1.* Intermediate and cumulative entropy across training epochs for CIFAR10 poisoned by Blend with using a ResNet18 model. Note, that entropy values of all training samples are rescaled to $[0, 1]$ at each epoch.

tasks (the primary functionality and the backdoor) formed by mixing two datasets of benign and poisonous samples, respectively. By retrieving the coreset of the primary task, we effectively mitigate the backdoor *and* maintain the natural performance. Moreover, focusing on the benign coreset explicitly does not attempt to retrieve all benign samples, thus, reducing the risk of accidentally including poisonous samples. Meanwhile, learning a model on this coreset is faster than on the full dataset, so that the overall runtime of the defense is comparable to a naive training run.

Coreset selection most commonly measures either *representativeness* (a data-centric criterion) or *informativeness* (a model-centric criterion). The former assesses the geometric coverage of data samples within the full dataset (Welling, 2009; Sener & Savarese, 2018). However, ensuring the full coverage increases the likelihood of including poisonous samples, e.g., using k-Center Greedy (Hekmatfar & Farahani, 2009) as shown in Figure 9. In contrast, informativeness measures a data sample's ability to increase model certainty (Huang et al., 2010). Interestingly, poisonous samples typically induce fast and stable convergence of the training loss (Li et al., 2021b; Zhao & Wressnegger, 2025), resulting in high confidence in them. This effect can thus also be seen in the entropy trend across every intermediate training epoch as shown in Figure 1 (left). The model exhibits a rapid uncertainty decrease for poisonous samples so that additional training on them does not increase prediction certainty, indicating their low informativeness to the model.

Therefore, selecting a coreset of the poisoned training dataset using the informativeness—without even considering the presence of two separate tasks—naturally puts stronger focus on the primary task. We find that such a phenomenon not only exist in uncertainty-based selection criteria (Coleman et al., 2020) but particularly in cumulative metrics over multiple training epochs (Toneva et al., 2019; Paul et al., 2021). Such methods can partially mitigate backdoors during training already, but fail to exclude all poisonous samples due to their inherent randomness of sampling at intermediate epochs and the selection process

itself. We thus use *Cumulative Entropy (*CENT*)* as a selection criterion that computes an uncertainty score for each sample by accumulating its entropy over multiple training epochs, capturing both predictive uncertainty and temporal consistency (cf. Figure 1, right). For backdoor mitigation, we then select samples with high cumulative entropy as the clean, informative coreset to use them for training a backdoor-free model.

**Contributions.** We present a novel interpretation of training-time backdoor defenses as a coreset selection problem, introducing *"Anti-backdoor Coreset Selection."* We implement this scheme based on a simple yet powerful criterion, the Cumulative Entropy, designed by us to reliably extract informative samples while ensuring to not misclassify poisonous samples during coreset selection. Extensive experiments across various backdoor attacks demonstrate the robustness of ABCS and the striking improvement over related work. We mitigate all investigated backdoor, achieve natural performance comparable to training on the original (clean) dataset, and also match the training time.

## 2. Threat Model

We consider backdoors that are introduced via data poisoning, that is, the adversaries manipulate samples from a training dataset $\mathcal{D}$. However, they cannot influence the training process. Naively training on the manipulated/poisoned dataset $\tilde{\mathcal{D}}$ learns the "primary task" (e.g., image classification) but also implants the backdoor as a "secondary task." More formally, given a training dataset $\mathcal{D} = \{(\mathbf{x}_i, y_i)\}_{i=1}^{N}$ with $N$ samples $\mathbf{x}_i \in \mathbb{R}^d$ and ground-truth labels $y_i \in \{0, 1, \ldots C-1\}$, where $C$ is the number of classes, the adversary poisons $N_p$ samples of $\mathcal{D}$ ($N_p \ll N$), resulting in a poisoned dataset $\tilde{\mathcal{D}} = \tilde{\mathcal{D}}_p \cup \mathcal{D}_c$ composed of a poisoned subset $\tilde{\mathcal{D}}_p$ and a clean subset $\mathcal{D}_c$, with the same size as $\mathcal{D}$, i.e., $|\mathcal{D}| = |\tilde{\mathcal{D}}|$. The poisoning rate of $\tilde{\mathcal{D}}$ thus is $\rho = N_p/N$.

Defenses using "dataset splitting" (Huang et al., 2022a; Gao et al., 2023; Zhu et al., 2023b; Zhao & Wressnegger, 2025; Chen et al., 2022) separate the already poisoned dataset $\tilde{\mathcal{D}}$ into a poisoned set $\mathcal{D}_{bad}$ and a benign set $\mathcal{D}_{bng}$, and use the latter to train a backdoor-free model. To maintain the natural performance, prior works (Gao et al., 2023; Zhu et al., 2023b; Zhao & Wressnegger, 2025; Chen et al., 2022) aim for $\mathcal{D}_{bng}$ containing as few poisonous samples as possible, ideally $\mathcal{D}_{bng} = \mathcal{D}_c$. We follow a similiar yet different aim: $\mathcal{D}_{bng}$ is intended to cover sufficient information of the clean subset $\mathcal{D}_c$ instead of include all clean samples. Consequently, it may be smaller than the clean subset, allowing $|\mathcal{D}_{bng}| < |\mathcal{D}_c|$ and thus yielding a selection ratio $r_{se} = |\mathcal{D}_{bng}|/|\tilde{\mathcal{D}}| \ll 1$. Moreover, the selected $\mathcal{D}_{bng}$ should contain almost no poisonous samples, i.e., $\rho_{bng} = \frac{|\mathcal{D}_{bng} \cap \tilde{\mathcal{D}}_p|}{|\mathcal{D}_{bng}|} \approx 0.0$.

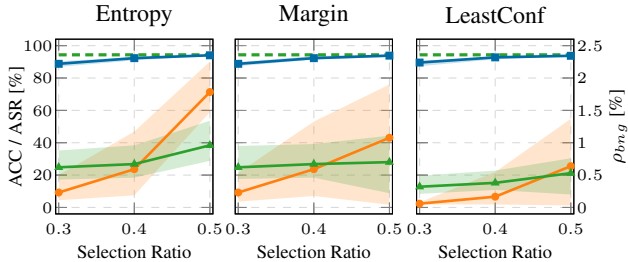

*(a)* Uncertainty-based Criteria

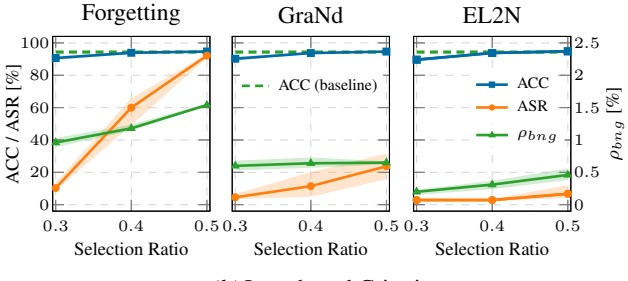

*(b)* Loss-based Criteria

*Figure 2.* Evaluation of coreset methods under the Blend attack using ResNet18 on CIFAR10 with $\rho = 5\%$. Error bars show the value range across five random runs per coreset size.

## 3. Coreset Selection as Backdoor Defense

Prior training-time defenses (Zhang et al., 2023; Zhu et al., 2023b) use the training loss for identifying benign and poisonous samples. There, the loss essentially serves as a proxy metric for the prediction confidence—or put differently, the reference model's (un)certainty about a sample. The model is certain about poisonous samples but uncertain about benign, hard-to-learn samples. Interestingly, for coreset selection, the community has explored loss-based criteria (Toneva et al., 2019; Paul et al., 2021) *but also* uncertainty measures directly (Coleman et al., 2020).

In the following, we thus specifically investigate these two categories of selection criteria (Section 3.1) and discuss their stability in the backdoor defense setting (Section 3.2). In Appendix A, we additionally investigate other selection criteria, such as, geometric-based (Hekmatfar & Farahani, 2009) or decision boundary based (Ducoffe & Precioso, 2018) criteria. Note that we discard balanced selection across classes to lower the likelihood of including poisonous samples from backdoor target class.

### 3.1. Uncertainty-Based and Loss-Based Selection

We examine the performance of uncertainty-based and loss-based coreset selection from a poisoned dataset $\tilde{\mathcal{D}}$. In line with the intuition on the relation between learning speed (Li et al., 2021b) and a model's uncertainty, we find that such criteria have a somewhat natural potential to mitigate backdooring attacks as can be seen in Figure 2 using the example

of the Blend attack (Chen et al., 2017).

Uncertainty-based approaches (Coleman et al., 2020) (Entropy, Margin, and LeastConf) consistently produce subsets, on which model training yields a natural performance comparable to the baseline of training on the clean dataset. At the same time, the portion of poisonous samples in these coresets remains below the default poisoning rate of $5\%$, indicating their low informativeness. Similarly, coresets selected by loss-based methods (Forgetting (Toneva et al., 2019), GraNd, and EL2N (Paul et al., 2021) shown in the bottom row) stably reproduce the natural accuracy. Note that, the variance (indicated as error bars) is significantly lower for loss-based criteria that either compute scores across multiple training epochs (Forgetting, EL2N) or uses random initialization (GraNd), taming the randomness associated with single training run.

Only, EL2N seems robust against Blend attack, yielding a moderate attack success rate (ASR) of 2–6 % and reaching high natural accuracy. It computes the $l_2$ norm between the predicted confidence of all classes and the one-hot encoded, ground-truth label. This way, EL2N essentially accumulates the mean squared error over training epochs. Figure 3 shows the vanilla (cumulative) version of EL2N and its variant of using the $l_2$ norm score per epoch. Clearly, the accumulation is crucial, although a relevant portion of the coreset remains poisoned.

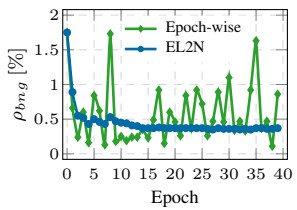

*Figure 3.* EL2N and its variant of sampling per epoch.

### 3.2. Instability of Uncertainty-Based Sampling

Uncertainty-based metrics are influenced by the randomness of intermediate sampling towards the end of the training, resulting in a large variance in both ASR and $\rho_{bng}$. Loss-based approaches are more stable, though. Thus, one-step sampling is not reliable for backdoor defense.

Based on the insights gained from EL2N and the smoothing effect observed in Figure 3, we set out to investigate whether accumulation can help to contain the variance of uncertainty-based metrics also. Figure 4 compares sampling per epoch and accumulating scores across epochs for selection criteria Entropy, Margin and Least Confidence (Coleman et al., 2020). For each, one can clearly see that although they occasionally achieve a low poisoning rate $\rho_{bng}$, more often than not the coreset contains a large number of poisonous samples. Interestingly, the issue becomes increasingly severe toward the end of the training, as the uncertainty gap between benign and poisonous samples diminishes significantly, making them less distinguishable.

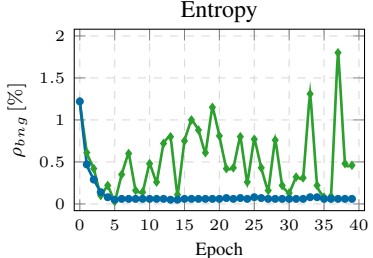 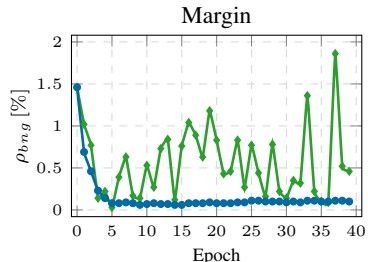 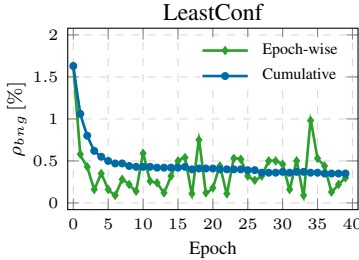

*Figure 4.* Epoch-wise sampling vs. accumulation for uncertainty-based coreset selection. Selection ratio is $0.4$. Each run trains a ResNet18 on CIFAR10 dataset that is poisoned by Blend attack with $\rho = 5\%$.

Accumulating uncertainty measurements across epochs, in turn, yields a robust and stable coreset selection, *and* simultaneously enables the elimination of poisonous samples. Unlike the accumulation of Entropy and Margin that yield near-zero $\rho_{bng}$, the accumulation of LeastConf cannot effectively eliminate poisonous samples, leading to a high $\rho_{bng}$.

**Backdoor Defense.** Next, we investigate the performance of the cumulative versions of the three uncertainty-based criteria when used for backdoor defense verbatim and summarize the results in Figure 5. Despite prior reports that plain Entropy is less performant in sampling an informative coreset (Guo et al., 2022), *Cumulative* Entropy yields higher natural accuracy than all other criteria. Moreover, the Cumulative Entropy consistently yields the smallest poisoning rate $\rho_{bng}$ and lowest ASR across coreset sizes, thus, making it an ideal choice for backdoor defense.

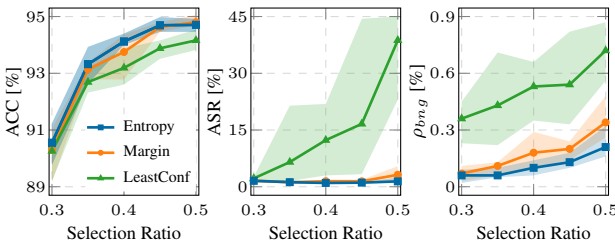

*Figure 5.* Training baseline model ResNet18 from scratch on coresets of CIFAR10 selected from a poisoned dataset with Blend attack by using different uncertainty criteria in the accumulation.

# 4. Anti-Backdoor Coreset Selection via Cumulative Entropy

Based on the observations made in the previous section, we propose a training-time defense, ABCS, using coreset selection via the Cumulative Entropy (CENT) criterion.

CENT is a simple yet effective coreset selection criterion, that robustly sorts data samples according to their informative value. It assigns low and high uncertainty to poisonous and benign samples, respectively, allowing to straightfor-

wardly choose a benign and informative subset. Additionally, it is beneficial to min-max-scale the scores of accumulated entropy across all data samples for each of the $T_{se}$ epochs, $\text{normalize}(a_i) := \frac{a_i - a_{\min}}{a_{\max} - a_{\min}}$, to avoid the interference in-between epochs, thus, further facilitating the stability of the overall metric. An empirical evidence in Table 5 demonstrate the benefit of using min-max normalization for the anti-backdoor coreset selection by CENT.

Formally, the CENT criterion to an input sample $\mathbf{x}_i$, i.e., CENT $(\mathbf{x}_i)$, can be calculated via:

$$\frac{1}{T_{se}} \sum_{t=1}^{T_{se}} \text{normalize} \underbrace{\left( \sum_{c=0}^{C-1} -p_{\theta_t}\left(c|\mathbf{x}_i\right) \cdot \log\left(p_{\theta_t}\left(c|\mathbf{x}_i\right)\right) \right)}_{\text{Normalized Shannon Entropy } H(\mathbf{x}_i)},$$

where accumulation over $T_{se}$ epochs and normalization across all sample in each is done as described in Section 3.2. However, *plainly using* CENT *for backdoor defense is not sufficient for complex backdooring attacks (Nguyen & Tran, 2021; Zeng et al., 2021; Qi et al., 2023).*

**Our defense, ABCS, thus comprises three phases:** First, a warm-up phase to stabilized the initial selection (Section 4.1). Second, the actual coreset selection with three individual steps (Section 4.2), and finally training a clean model from scratch using the selected coreset (Section 4.3).

## 4.1. Warm-Up

Each backdooring attack has a varying learning difficulty (Zhao & Wressnegger, 2024), so that we first bootstrap training in a warm-up phase of $T_{wa}$ epochs, yielding an initial model. We use this model to determine the size $s$ of the coreset to be selected as $\mathcal{D}_{bng}$, which in turn specifies the selection ratio $r_{se}$ for a poisoned dataset $\tilde{\mathcal{D}}$ as $r_{se} = s/|\tilde{\mathcal{D}}|$. Many approaches (Toneva et al., 2019; Qin et al., 2024) use the mean of the importance score as a threshold to determine the coreset size. However, wrongly predicted samples are considered as not learned by the model despite the uncertainty (Toneva et al., 2019). Given the number of correct predictions $N_{cr} = \sum_{i=1}^{N} \mathbb{1}_{\arg\max_c(f_\theta(\mathbf{x}_i)_c)=y_i}$, hence, the

cut-off threshold $\tau$ of $\text{CENT}\,(\mathbf{x}_i)$ is defined as:

$$\tau = \frac{1}{T_{wa}} \sum_{t=1}^{T_{wa}} \left( \frac{1}{N_{cr}} \sum_{i=1}^{N} H\,(\mathbf{x}_i) \cdot \mathbb{1}_{\arg\max_c\left(f_{\theta_t}(\mathbf{x}_i)_c\right) = y_i} \right)$$

and thus $\quad s = \sum_{i=1}^{N} \mathbb{1}_{\text{CENT}(\mathbf{x}_i) > \tau}$ , which determines the final coreset size for the next selection stage, i.e., $|\mathcal{D}_{bng}| = s$.

### 4.2. Selecting the Coreset

After the warm-up stage, we resume to train the model for $T_{se}$ epochs. Each epoch serves three purposes: First, learning the entropy distribution of all samples in $\tilde{\mathcal{D}}$; Second, disentangling benign and uncertain samples from poisoned ones; Third, calculating the normalized entropy of individual samples and accumulate it with results from the previous epoch, yielding an updated $\text{CENT}$ at the intermediate epoch. The coreset selection procedure within each individual epoch is thus implemented in three steps:

**Step-1. Learning the entire dataset $\tilde{\mathcal{D}}$ .** Backdooring attacks do not influence the natural performance and implicitly preserve a data distribution comparable to training on the clean dataset. Thus, we first train on the entire poisoned dataset $\tilde{\mathcal{D}}$ for one epoch to obtain a model $\theta^*$, which outputs an intermediate entropy distribution of the entire dataset.

**Step-2. Unlearning uncertain samples.** Cumulative Entropy alone does not guarantee a coreset free of poisonous samples of various backdoors, particularly for stealthier ones, e.g., WaNet (cf. Figure 7). We thus unlearn uncertain samples to enlarge the discrepancy in uncertainty between poisonous and benign samples, thereby reducing the overlap between the final coreset $\mathcal{D}_{bng}$ and the poisonous samples $\tilde{\mathcal{D}}_p$. To do so, we first measure the entropy of each sample $\mathbf{x}_i$, i.e., $H(\mathbf{x}_i)$, at intermediate epoch $t$ and select samples larger than the average as the intermediate unlearning subset $\mathcal{D}_{ul} = \left\{ (\mathbf{x}_i, y_i) \mid H\,(\mathbf{x}_i) > \frac{1}{N_{cr}} \sum_{i=1}^{N} H\,(\mathbf{x}_i) \cdot \mathbb{1}_{\arg\max_c\left(f_{\theta_t}(\mathbf{x}_i)_c\right) = y_i} \right\}$

Additionally, we use label smoothing (Müller et al., 2019) to counter instability during unlearning (Di et al., 2026; Fan et al., 2024; Neel et al., 2020) and boost the forgetting of the unlearning set. The label smoothing pushes the prediction still towards ground-truth but in a more smooth distribution, implicitly enlarging the uncertainty of hard-to-learn, benign samples, making them distant to poisonous samples in $\text{CENT}$ distribution. Given a smoothing factor $\varepsilon$, the smoothed target output $\mathbf{z}$ (often also referred to as "smoothed/soft label" in related work) of sample $\mathbf{x}$ is thus

expressed as:

$$z_c = \begin{cases} 1 - \varepsilon + \frac{\varepsilon}{C} & \text{if } c = y \\ \frac{\varepsilon}{C} & \text{otherwise.} \end{cases}$$

A large $\varepsilon$ lowers the targeted confidence of the sample's label $y$, implicitly enlarging the prediction uncertainty. An ablation study of smoothing factor $\varepsilon$ is shown in Section 5.3.

Moreover, we include $l_2$ regularization based on the current and the previous models' weights, $\theta$ and $\theta^*$, respectively, to restrict the overall change/impact on the learned task. Hence, we use the following loss function to unlearn samples from $\mathcal{D}_{ul}$:

$$\mathcal{L}_{\text{UL}}(\mathbf{x}, \mathbf{z}, \theta) := \gamma \cdot \overbrace{\mathcal{L}_{CE}\,(\mathbf{x}, \mathbf{z}, \theta)}^{\text{label smoothing}} + \overbrace{\sum_j (\theta_j - \theta_j^*)^2}^{l_2 \text{ regularization}},$$

where $\gamma$ is a regularization parameter. To avoid model collapse, we additionally lower the learning rate $lr$ to $10\,\%$ of the initial value. Different from the unlearning by gradient ascending (Li et al., 2021b), which leads to an extraordinarily large learning divergence, label smoothing stabilizes unlearning because the "smoothed label" still specifies a fixed target where the ground-truth label has the highest score and, thus, unlearning will not push the prediction to an extreme case. Additionally, label smoothing implicitly increases the entropy values of samples in $\mathcal{D}_{ul}$.

**Step-3. Updating the cumulative entropy.** After Step-2, we compute the normalized entropy $H(\mathbf{x}_i)$ for every sample $\mathbf{x}_i$. For $t = 1$, $\text{CENT}(\mathbf{x}_i)$ is directly identical to $H(\mathbf{x}_i)$. For $t > 1$, the newly computed $H(\mathbf{x}_i)$ is accumulated to the cumulative entropy values obtained from previous epochs $t = \{1, \ldots, t-1\}$, yielding an updated $\text{CENT}(\mathbf{x}_i)$.

Across $T_{se}$ epochs, $\text{CENT}(\mathbf{x}_i)$ progressively aligns with the data's underlying distribution of informativeness. Meanwhile, as the model has high confidence on the poisonous samples (yielding low uncertainty), the discrepancy between poisonous samples and informative clean samples is significantly enlarged in the view of Cumulative Entropy. After running the last selection epoch (i.e., $t = T_{se}$), we sort all training samples by their $\text{CENT}$ scores and select $s$ samples (the coreset size determined in Section 4.1) with highest $\text{CENT}$ values, forming the final clean coreset $\mathcal{D}_{bng}$.

### 4.3. Final Training

Finally, we train a model on the final coreset $\mathcal{D}_{bng}$ from scratch for $T_{cln}$ epochs using regular cross-entropy loss: $\arg\min_\theta \sum_{(\mathbf{x}, y) \in \mathcal{D}_{bng}} \mathcal{L}_{\text{CE}}\,(\mathbf{x}, y, \theta)$. As the coreset $\mathcal{D}_{bng}$ is significantly smaller than the full training dataset $\tilde{\mathcal{D}}$, training the final model is fast. Thus, ABCS's overall training time (including coreset selection and training the model) is comparable to naive (unprotected) learning.

*Table 1.* Comparing ABCS to prior training-time defenses. Each defense experiment contains the averaged value and the standard deviation in [%] across five random runs. The best results across all defenses are highlighted as **boldface**. The defense failure (i.e., ASR > 50 %) is shown as **orange boldface**. We discard ASR for No-Defense as all attacks reach a ASR in the range of 95 % to 100 %.

*(a)* CIFAR10

| Attack | No-Defense | ABL | | | DBD | | | CBD | | |
|---|---|---|---|---|---|---|---|---|---|---|
| | ACC (↑) | ACC (↑) | ASR (↓) | DER (↑) | ACC (↑) | ASR (↓) | DER (↑) | ACC (↑) | ASR (↓) | DER (↑) |
| BadNets | 94.53±0.24 | 92.96±0.35 | **0.36**±0.06 | 99.03±0.18 | 92.19±1.69 | 2.24±0.22 | 97.71±0.88 | 88.63±1.53 | 20.15±42.28 | 86.98±21.80 |
| Blend | 94.49±0.09 | 88.29±2.63 | 34.98±38.17 | 78.05±20.26 | 93.12±0.50 | 7.25±3.20 | 94.33±1.79 | 89.41±1.84 | 47.21±27.66 | 72.49±13.53 |
| CLB | 94.46±0.13 | 85.75±1.67 | 1.46±1.36 | 94.92±0.45 | 91.85±1.15 | 3.85±6.24 | 96.77±2.66 | 89.11±1.84 | 14.71±32.90 | 89.97±15.60 |
| IAB | 94.53±0.06 | 92.59±0.48 | 3.10±0.89 | 97.40±0.36 | 92.34±4.99 | **99.82**±0.13 | 48.47±1.85 | 89.84±0.94 | 8.40±2.82 | 93.38±1.81 |
| WaNet | 94.02±0.19 | 88.64±2.41 | **59.99**±48.83 | 65.50±25.44 | 91.97±0.62 | 2.42±0.29 | 95.77±0.44 | 90.05±2.06 | 48.19±42.67 | 71.92±21.54 |
| ISSBA | 94.51±0.22 | 87.54±1.77 | 0.98±0.32 | 96.02±0.77 | 92.53±0.43 | 10.91±1.13 | 93.55±0.42 | 93.01±0.27 | **83.08**±34.68 | 57.71±17.31 |
| LF | 94.70±0.08 | 89.38±3.38 | 41.42±51.18 | 75.80±25.36 | 92.86±0.40 | **96.69**±1.93 | 49.94±1.11 | 90.85±0.59 | **69.37**±27.15 | 62.55±13.59 |
| A-Blend | 94.02±0.06 | 83.87±3.81 | **62.35**±9.65 | 61.74±6.17 | 92.78±0.73 | 6.06±1.59 | 94.34±0.73 | 92.26±0.80 | **60.36**±7.91 | 66.93±3.57 |
| Average | 94.41 | 88.63 | 25.58 | 83.56 | 92.45 | 28.66 | 83.86 | 90.40 | 43.94 | 75.24 |
| WorstCase | 94.02 | 83.87 | **62.35** | 61.74 | 91.85 | **99.82** | 48.47 | 88.63 | **83.08** | 57.71 |

| Attack | No-Defense | V&B | | | Harvey | | | ABCS (Ours) | | |
|---|---|---|---|---|---|---|---|---|---|---|
| | ACC (↑) | ACC (↑) | ASR (↓) | DER (↑) | ACC (↑) | ASR (↓) | DER (↑) | ACC (↑) | ASR (↓) | DER (↑) |
| BadNets | 94.53±0.24 | **94.64**±0.16 | 0.91±0.13 | **99.53**±0.05 | 93.92±0.31 | 1.08±0.40 | 99.16±0.23 | 94.38±0.23 | 1.00±0.17 | 99.42±0.19 |
| Blend | 94.49±0.09 | 94.07±0.37 | 0.94±0.06 | 97.96±0.18 | 94.05±0.16 | 0.92±0.32 | 97.96±0.20 | **94.62**±0.36 | 0.77±0.08 | **98.22**±0.07 |
| CLB | 94.46±0.13 | 94.23±0.11 | 1.06±0.34 | 99.36±0.17 | 93.86±0.22 | **0.45**±0.08 | 99.47±0.09 | **94.51**±0.23 | 0.88±0.10 | **99.54**±0.06 |
| IAB | 94.53±0.06 | 94.14±0.72 | 13.85±28.85 | 92.79±14.78 | 93.54±0.53 | 2.01±0.74 | 98.42±0.49 | **94.30**±0.16 | **1.22**±0.37 | **99.19**±0.19 |
| WaNet | 94.02±0.19 | **94.33**±0.13 | 1.10±0.39 | 97.45±0.19 | 93.51±0.20 | 1.79±0.93 | 96.85±0.55 | 94.13±0.22 | **0.89**±0.16 | **97.53**±0.13 |
| ISSBA | 94.51±0.22 | 93.87±0.77 | **0.64**±0.63 | **99.36**±0.62 | 93.35±0.48 | 1.26±0.26 | 98.79±0.36 | **94.66**±0.41 | 1.22±0.09 | 99.33±0.15 |
| LF | 94.70±0.08 | 93.88±1.28 | 57.34±41.79 | 70.15±21.09 | 93.44±0.57 | 39.63±50.98 | 78.75±25.19 | **94.06**±0.30 | **3.04**±0.55 | **97.33**±0.17 |
| A-Blend | 94.02±0.06 | 92.16±0.78 | **96.03**±5.43 | 50.03±2.01 | 93.63±0.30 | **75.42**±3.90 | 60.08±1.85 | **94.37**±0.27 | **5.71**±1.59 | **95.12**±0.77 |
| Average | 94.41 | 93.92 | 21.48 | 88.33 | 93.66 | 15.32 | 91.18 | **94.38** | **1.84** | **98.21** |
| WorstCase | 94.02 | 92.16 | **96.03** | 50.03 | 93.35 | **75.42** | 60.08 | **94.06** | **5.71** | **95.12** |

*(b)* Tiny-ImageNet

| Attack | No-Defense | ABL | | | DBD | | | CBD | | |
|---|---|---|---|---|---|---|---|---|---|---|
| | ACC (↑) | ACC (↑) | ASR (↓) | DER (↑) | ACC (↑) | ASR (↓) | DER (↑) | ACC (↑) | ASR (↓) | DER (↑) |
| BadNets | 61.77±0.17 | 47.14±1.22 | **0.00**±0.00 | 92.66±0.61 | 51.31±1.36 | **98.93**±1.69 | 45.29±1.50 | 48.93±0.68 | 0.11±0.04 | 93.50±0.32 |
| Blend | 61.44±0.37 | 38.40±1.16 | **0.00**±0.00 | 88.30±0.58 | 51.68±0.39 | **99.87**±0.21 | 45.12±0.20 | 49.11±0.58 | 16.51±22.55 | 85.40±11.52 |
| IAB | 61.98±0.10 | 44.73±0.54 | **0.00**±0.00 | 91.37±0.27 | 50.82±0.19 | **99.86**±0.19 | 44.49±0.08 | 49.40±0.69 | 0.31±0.06 | 93.55±0.36 |
| WaNet | 60.71±0.02 | 41.88±5.43 | **94.82**±4.78 | 42.66±3.51 | 50.99±0.13 | **99.41**±1.02 | 45.26±0.15 | 47.89±0.78 | **60.81**±18.96 | 62.68±9.61 |
| ISSBA | 61.29±0.12 | 35.38±0.73 | **0.07**±0.10 | 86.99±0.42 | 51.04±0.27 | **96.81**±1.39 | 46.45±0.57 | 44.90±0.16 | 0.50±0.27 | 91.54±0.07 |
| LF | 61.40±0.11 | 38.59±0.83 | **0.00**±0.00 | 87.51±0.42 | 51.00±0.14 | **96.04**±0.75 | 45.70±0.38 | 48.74±0.10 | 7.95±5.18 | 88.61±2.64 |
| A-Blend | 60.97±0.06 | 39.20±0.97 | **76.07**±13.73 | 49.56±7.04 | 50.64±0.21 | **98.69**±1.23 | 44.84±0.11 | 48.34±1.15 | **88.32**±11.91 | 48.21±5.71 |
| Average | 61.37 | 40.76 | 24.42 | 77.01 | 51.07 | **98.52** | 45.31 | 48.19 | 24.93 | 80.50 |
| WorstCase | 60.71 | 35.38 | **94.82** | 42.66 | 50.64 | **99.87** | 44.49 | 44.90 | **88.32** | 48.21 |

| Attack | No-Defense | V&B | | | Harvey | | | ABCS (Ours) | | |
|---|---|---|---|---|---|---|---|---|---|---|
| | ACC (↑) | ACC (↑) | ASR (↓) | DER (↑) | ACC (↑) | ASR (↓) | DER (↑) | ACC (↑) | ASR (↓) | DER (↑) |
| BadNets | 61.77±0.17 | 61.35±0.16 | 0.51±0.15 | 99.51±0.08 | **61.86**±0.56 | 0.02±0.02 | **99.88**±0.15 | 61.24±0.24 | 0.11±0.04 | 99.65±0.10 |
| Blend | 61.44±0.37 | **61.84**±0.28 | 2.41±2.35 | 98.61±1.17 | 61.28±0.58 | 0.18±0.13 | **99.57**±0.21 | 61.11±0.07 | 0.18±0.05 | 99.56±0.06 |
| IAB | 61.98±0.10 | 61.64±0.24 | 0.15±0.04 | **99.75**±0.11 | 61.26±0.11 | 3.55±2.07 | 97.86±1.09 | **61.68**±0.44 | 0.22±0.04 | 99.73±0.21 |
| WaNet | 60.71±0.02 | 60.79±0.32 | 1.65±1.12 | 98.62±0.56 | **61.08**±0.48 | 1.91±2.06 | 98.53±1.05 | 60.54±0.78 | **0.51**±0.22 | **99.06**±0.24 |
| ISSBA | 61.29±0.12 | **58.02**±0.68 | 14.34±6.26 | 91.18±2.82 | 57.95±0.75 | 0.39±0.21 | 98.12±0.30 | 58.01±0.37 | 0.25±0.05 | **98.22**±0.16 |
| LF | 61.40±0.11 | **61.91**±0.63 | 17.87±13.74 | 89.98±6.87 | 61.54±0.99 | **79.33**±17.97 | 59.21±8.76 | 61.44±0.35 | 0.18±0.15 | **98.77**±0.02 |
| A-Blend | 60.97±0.06 | 56.85±0.12 | **73.13**±16.42 | 59.85±8.26 | **60.93**±0.60 | **74.26**±12.08 | 61.23±6.10 | 60.03±0.50 | 1.15±0.43 | 97.43±0.25 |
| Average | 61.37 | 60.34 | 15.72 | 91.07 | **60.84** | 22.80 | 87.77 | 60.58 | **0.37** | **98.92** |
| WorstCase | 60.71 | 56.85 | **73.13** | 59.85 | 57.95 | **79.33** | 59.21 | **58.01** | **1.15** | **97.43** |

# 5. Evaluation

We conduct extensive experiments across two small-scale datasets CIFAR10 (Krizhevsky et al., 2008) and GT-SRB (Stallkamp et al., 2012) with ResNet18 (He et al., 2016) , and one large-scale dataset Tiny-ImageNet (Le & Yang, 2015) with ResNet34 (He et al., 2016). To account for randomness, each experiment is repeated with five random seeds on CIFAR10 and GTSRB, and three seeds on Tiny-ImageNet. In the appendix, we provide details of experimental setups (cf. Appendix C), extend the evaluation (cf. Appendix D) to experiments for GTSRB, ablation studies on ABCS' hyper-settings, the robustness analysis across diverse attack settings, experiments for all-to-all attacks and for text classification task, and comparison to one defense using reference clean data, test ABCS against four adaptive attacks (cf. Appendix E), and finally analyze the coverage of selected coresets to full datasets (cf. Appendix F).

*Table 2.* Comparing ABCS to training-time defenses on clean (not poisoned) datasets.

| Dataset | Naive | ABL | DBD | CBD | V&B | Harvey | ABCS |
|---------|-------|-----|-----|-----|-----|--------|------|
| CIFAR10 | $94.64_{\pm 0.17}$ | $83.42_{\pm 3.35}$ | $92.32_{\pm 2.23}$ | $92.90_{\pm 0.41}$ | $94.16_{\pm 0.41}$ | $93.80_{\pm 0.29}$ | $\mathbf{94.77_{\pm 0.11}}$ |
| GTSRB | $98.36_{\pm 0.21}$ | $91.79_{\pm 2.11}$ | $93.38_{\pm 0.37}$ | $80.54_{\pm 9.73}$ | $92.50_{\pm 2.44}$ | $97.95_{\pm 0.22}$ | $\mathbf{98.12_{\pm 0.28}}$ |
| Tiny-ImageNet | $62.22_{\pm 0.45}$ | $37.46_{\pm 1.86}$ | $51.81_{\pm 0.67}$ | $51.00_{\pm 0.64}$ | $58.87_{\pm 0.76}$ | $60.86_{\pm 0.45}$ | $\mathbf{61.19_{\pm 0.26}}$ |

**Attacks and Defenses.** We consider eight dataset poisoning backdoors: BadNets (Gu et al., 2017), Blend (Chen et al., 2017), CLB (Turner et al., 2019), IAB (Nguyen & Tran, 2020), WaNet (Nguyen & Tran, 2021), ISSBA (Li et al., 2021a), Low Frequency (LF; Zeng et al., 2021), and Adaptive Blend (A-Blend; Qi et al., 2023). For a fair comparison, we follow the default implementation of each attack. We choose class 0 as target label and use a $5\%$ poisoning rate, except for CLB, which uses $\rho = 50\%$ on the target class. For Tiny-ImageNet, CLB is excluded due to its ineffectiveness. In the evaluation, we compare ABCS to six state-of-the-art training-time defenses that require no prior clean reference data: ABL (Li et al., 2021b), DBD (Huang et al., 2022a), CBD (Zhang et al., 2023), V&B (Zhu et al., 2023b), and Harvey (Zhao & Wressnegger, 2025). We evaluate each defense using its default settings and hyperparameters. For ABCS, we set $T_{wa} = 10$ and $T_{se} = 40$. The unlearning step during coreset selection uses $\varepsilon = 0.9$, and $\gamma = 0.1$ and $0.01$ for small-scale and for large-scale datasets. Moreover, we use ADAM optimizer with learning rate $0.001$ during warm-up and coreset selection.

**Evaluation metrics.** We present the defensive performance with three metrics: Natural Accuracy (ACC), Attack Success Rate (ASR), and Defense Effectiveness Rate (DER) (Zhu et al., 2023a). Formally, DER is defined as: $\text{DER} = \left[\max\left(0, \Delta \text{ASR}\right) - \max\left(0, \Delta \text{ACC}\right) + 1\right]/2$, where $\Delta \text{ASR}$ is the drop of ASR and $\Delta \text{ACC}$ is the drop of ACC. The optimal defense achieves DER equal $100\%$.

### 5.1. Performance of Training-Time Defenses

Table 1 shows the results on CIFAR10 and Tiny-ImageNet. Across five random trials, prior defenses struggle to balance natural accuracy and the reduction of ASR. More notably, each is bypassed by one or multiple attacks. Specifically, ABL and CBD exhibit high variance in ASR while simultaneously compromising natural accuracy. DBD shows stability against the randomness but fails under several attacks. Advanced methods like V&B and Harvey maintain high natural accuracy but remain vulnerable to strong attacks such as LF and A-Blend. In contrast, ABCS shows consistent robustness across attacks and random seeds, while preserving ACC comparable to No-Defense. Although ABCS does not always achieve the best results on individual poisoned datasets, its strong ASR reduction and ACC retention consistently yield a high DER—above $95\%$ on CIFAR10 and $97\%$ on Tiny-ImageNet—demonstrating its effectiveness in selecting informative coresets that exclude poisonous data.

**Adaptability to clean datasets.** Prior defenses primarily focus on mitigating backdoors. However, they often suffer a drop in natural accuracy compared to naive training, highlighting a key limitation in their adaptability to clean (not poisoned) datasets (cf. Table 2). Differently, ABCS leverages the high informativeness of coresets, enabling full adaptability when training on clean datasets.

**Time consumption.** ABCS benefits from the small coreset size, resulting in faster training comparable to native training on Tiny-ImageNet (cf. Table 3), whereas other methods require more time, in particular for DBD defense.

*Table 3.* Time consumption of native training and all defenses.

| Dataset | Naive (h) | ABL | DBD | CBD | V&B | Harvey | ABCS |
|---------|-----------|-----|-----|-----|-----|--------|------|
| CIFAR10 | 1.47 | $\times 1.08$ | $\times 12.09$ | $\times 1.10$ | $\times 2.45$ | $\times 1.82$ | $\times 0.97$ |
| GTSRB | 0.86 | $\times 1.25$ | $\times 38.64$ | $\times 1.12$ | $\times 5.71$ | $\times 1.76$ | $\times 0.75$ |
| Tiny-ImageNet | 9.14 | $\times 1.17$ | $\times 13.92$ | $\times 1.20$ | $\times 5.58$ | $\times 1.82$ | $\times 1.25$ |

### 5.2. Investigation of Coreset Properties

We particularly look at the overall selection ratio of the coreset and its poisoning ratio, and finally inspect how well the class disribution is preserved.

**Poisoning ratio $\rho_{bng}$ and selection ratio $r_{se}$.** Unlike prior defenses (Huang et al., 2022a; Gao et al., 2023; Zhao & Wressnegger, 2025) that aim for precise dataset splitting, our method selects coresets smaller than the entire clean dataset (i.e., selection ratio $r_{se} \ll 100\%$). Table 4 summarizes the selected coresets for CIFAR10, where the selection ratio $r_{se}$ is around $55\%$ and the poisoning rate $\rho_{bng}$ is significantly reduced to near $0\%$. This effectively prevent backdoor implanting in the model training, and the strong natural performance shown in Table 1 underscores the high informativeness of the selected coresets by ABCS.

*Table 4.* Coresets of CIFAR10 under different attack scenarios.

| Attack | $r_{se}$ [%] | $\rho_{bng}$ [%] |
|--------|--------------|------------------|
| BadNets | $54.52_{\pm 0.58}$ | $0.00_{\pm 0.00}$ |
| Blend | $53.85_{\pm 0.58}$ | $0.00_{\pm 0.00}$ |
| CLB | $54.53_{\pm 0.45}$ | $0.00_{\pm 0.00}$ |
| IAB | $54.90_{\pm 0.83}$ | $0.01_{\pm 0.01}$ |
| WaNet | $57.83_{\pm 0.24}$ | $0.20_{\pm 0.09}$ |
| ISSBA | $54.04_{\pm 1.05}$ | $0.00_{\pm 0.00}$ |
| LF | $54.89_{\pm 0.77}$ | $0.19_{\pm 0.05}$ |
| A-Blend | $56.03_{\pm 0.51}$ | $0.54_{\pm 0.15}$ |

**Preservation of class-wise distribution.** Figure 6 shows the class-wise distribution of natural accuracy and data composition for CIFAR10. After training on coresets under

*Table 5.* Impact of min-max normalization in CENT criterion.

| Attack | Min-Max Norm. | $r_{se} = 0.3$ | | | $r_{se} = 0.4$ | | | $r_{se} = 0.5$ | | |
|---|---|---|---|---|---|---|---|---|---|---|
| | | ACC | ASR | $\rho_{bng}$ | ACC | ASR | $\rho_{bng}$ | ACC | ASR | $\rho_{bng}$ |
| Blend | ✗ | $90.82_{\pm 0.45}$ | $\mathbf{1.35}_{\pm 0.34}$ | $\mathbf{0.06}_{\pm 0.03}$ | $93.89_{\pm 0.24}$ | $1.12_{\pm 0.31}$ | $\mathbf{0.09}_{\pm 0.03}$ | $94.63_{\pm 0.37}$ | $1.41_{\pm 1.07}$ | $0.21_{\pm 0.07}$ |
| | ✓ | $\mathbf{90.89}_{\pm 0.20}$ | $1.38_{\pm 0.05}$ | $\mathbf{0.06}_{\pm 0.03}$ | $\mathbf{94.12}_{\pm 0.17}$ | $\mathbf{1.00}_{\pm 0.25}$ | $0.10_{\pm 0.03}$ | $\mathbf{94.71}_{\pm 0.18}$ | $\mathbf{1.22}_{\pm 0.20}$ | $\mathbf{0.19}_{\pm 0.03}$ |
| WaNet | ✗ | $88.12_{\pm 0.47}$ | $3.36_{\pm 0.33}$ | $1.50_{\pm 0.33}$ | $91.55_{\pm 0.93}$ | $1.55_{\pm 0.49}$ | $0.62_{\pm 0.19}$ | $93.94_{\pm 0.24}$ | $6.86_{\pm 5.29}$ | $1.36_{\pm 0.46}$ |
| | ✓ | $\mathbf{88.72}_{\pm 0.44}$ | $\mathbf{3.05}_{\pm 0.44}$ | $\mathbf{1.41}_{\pm 0.27}$ | $\mathbf{92.63}_{\pm 0.34}$ | $\mathbf{1.18}_{\pm 0.13}$ | $\mathbf{0.50}_{\pm 0.16}$ | $\mathbf{94.14}_{\pm 0.26}$ | $\mathbf{1.90}_{\pm 0.76}$ | $\mathbf{0.79}_{\pm 0.27}$ |

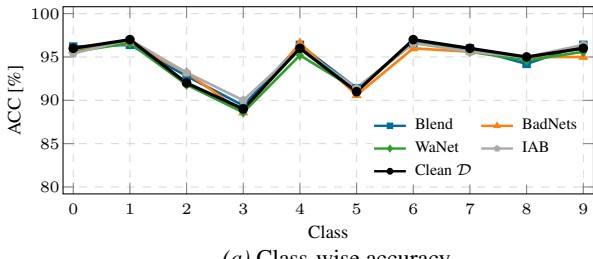

*(a)* Class-wise accuracy

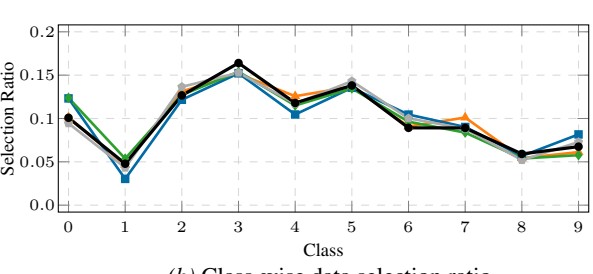

*(b)* Class-wise data selection ratio

*Figure 6.* Class-wise distribution on ResNet18 with CIFAR10 across different attacks over five runs. "Clean $\mathcal{D}$" refers to using the full dataset for both (a) model training and (b) coreset selection.

various attacks (left), the class-wise accuracy has a distribution similar to that obtained from the clean dataset $\mathcal{D}$. Also the class-wise composition of the coresets remain consistent across attacks (right). ABCS consistently extracts coresets that retain informativeness regardless of data poisoning and achieves high coverage of the full clean dataset (cf. Appendix F), thereby robustly preserving natural performance.

### 5.3. Ablation Study

Finally, we conduct an ablation study on different building blocks of our method, such as the warm-up and unlearning, the label smoothing factor $\varepsilon$, and CENT's normalization.

**Warm-up training and unlearning.** Using the Cumulative Entropy criterion can effectively mitigate strong neural backdoor attacks, such as the Blend attack (cf. Figure 5). The incorporation of warm-up and unlearning is particularly beneficial for stealthier attacks, where the convergence to the backdoor occurs more slowly. Figure 7 showcases coresets with fixed selection ratios against the WaNet attack, where the coreset selection by Cumulative Entropy alone cannot fully exclude poisonous samples. While the natural

accuracy remains unaffected, applying warm-up reduces the poisoning rate $\rho_{bng}$ of coresets, which is further lowered by executing the additional unlearning step.

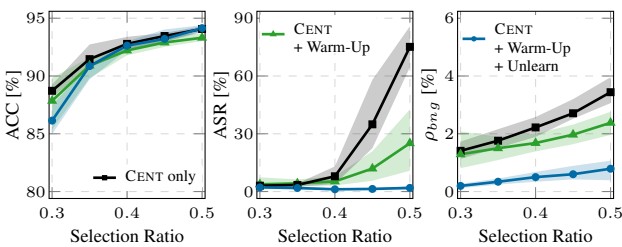

*Figure 7.* Comparing the impact of using warm-up and unlearning in ABCS. Baseline model ResNet18 is trained from scratch on selected coresets of CIFAR10 under WaNet attack.

**Label smoothing factor $\varepsilon$.** The value $\varepsilon$ is proportional to the strength of label smoothing. Thus, increasing $\varepsilon$ will amplify the unlearning effect, and vice versa. Figure 8 below investigates the impact of different $\varepsilon$ values on ABCS's defense. Varying the smoothing factor does not affect the natural performance represented by the selected coreset, making ACC consistently high. However, an overly small $\varepsilon$ indeed weakens the unlearning effect, finally resulting in an insufficient removal of poisoned samples, particularly under the WaNet attack. As the result, residual poisoned samples in the coreset lead to a higher ASR. Overall, the effectiveness of ABCS depends on choosing $\varepsilon$ within a reasonable range, rather than relying on a specific fixed value.

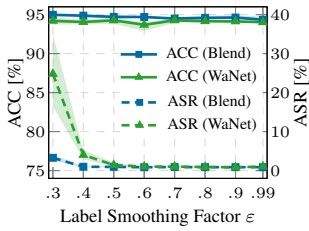

*Figure 8.* Ablation study on label smoothing factor $\varepsilon$.

**Min-max normalization in CENT.** The min-max normalization balances the entropy scale across epochs from early to late training time. Experiments on CIFAR10 (cf. Table 5) show that, despite low ASR and poisoning rates of coresets, using normalization improves ACC and enhances the elimination of poisonous samples particularly for WaNet.

# 6. Conclusion

Mitigating data-poisoning backdoors in the training procedure is challenging due to the high risk of misidentifying poisonous samples. Our approach, ABCS, leverages the defense nature of coreset selection and introduce the Cumulative Entropy criterion to extract an informative coreset while effectively excluding poisonous data. Training on this benign and informative coreset reproduces natural performance comparable to training on a fully clean dataset. ABCS demonstrates robustness across a wide range of attacks and diverse datasets. Notably, ABCS adapts well even in clean settings and incurs a computational cost similar to the naive training, reinforcing its practicality and giving rise to a promising novel paradigm in training-time backdoor defense: *Anti-Backdoor Coreset Selection*.

**Limitations.** While ABCS demonstrates strong empirical performance, it lacks a theoretical foundation. Future work could investigate latent space separation between coreset and poisonous samples, focusing on data informativeness and backdoor effectiveness. Compared to the optimal coreset for clean datasets, ABCS results in a larger coreset size, suggesting further exploration into optimal selection from poisoned datasets. Its defense against sophisticated attacks, e.g., A-Blend, remains suboptimal, as it does not reduce ASR to the minimum, indicating the need for improvement.

# Acknowledgments

We gratefully acknowledge funding by the Helmholtz Association (HGF) within topic "46.23 Engineering Secure Systems."

# Impact Statement

Deep neural networks are widely applied across numerous domains, which makes evaluating their security in real-world settings essential. In this work, we propose *Anti-Backdoor Coreset Selection*, a simple yet effective defense scheme for training a backdoor-free model from a poisoned dataset. Our approach is developed strictly from the perspective of a defender, as defined in the threat model. Therefore, this research does not introduce any ethical concerns or create additional security risks.

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

# Appendix

In the following, we first provide the coreset selection using other criteria (cf. Appendix A) and compare cumulative uncertainty criteria to three loss-based ones under WaNet attack (cf. Appendix B). In Appendix C, we elaborate on the experimental setups of considered attacks and defenses. After that, we proceed to analyze the performance of ABCS in Appendix D, including: the defense performance on GTSRB dataset (Appendix D.1), the ablation study on different hyper-settings of ABCS (Appendix D.2), the robustness under different target classes and poisoning rates (Appendix D.3), the evaluation across other model architectures (Appendix D.4), the resistance against all-to-all attacks (Appendix D.5), the effectiveness on the text classification task with using BERT models (Appendix D.6), and the comparison to the dataset splitting approach using the clean reference data (Appendix D.7) . Furthermore, we evaluate the resistance of ABCS against various adaptive adversaries that constructs poisoned datasets by intensionally enlarging the prediction uncertainty or/and the learning difficulty of poisoned samples (Appendix E). Finally, we investigate the geometric distribution of selected coresets in the latent space representation to empirically examine the coverage of the entire clean dataset (Appendix F).

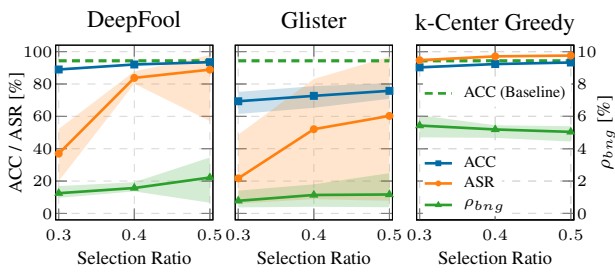

*Figure 9.* Coreset selected by three additional criteria on CIFAR10 poisoned by Blend with $\rho = 5\%$ using ResNet18. Error bars indicate the range across five random trials per coreset size.

## A. Analysis of Other Coreset Selection Criteria

In Figure 9, we investigate the defensive behavior of three selection criteria: DeepFool (Ducoffe & Precioso, 2018), Glister (Killamsetty et al., 2020) and k-Center Greedy (Hekmatfar & Farahani, 2009). Unlike uncertainty-based and

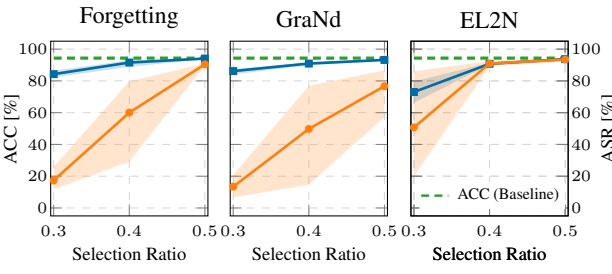

*(a)* Loss-based Criteria

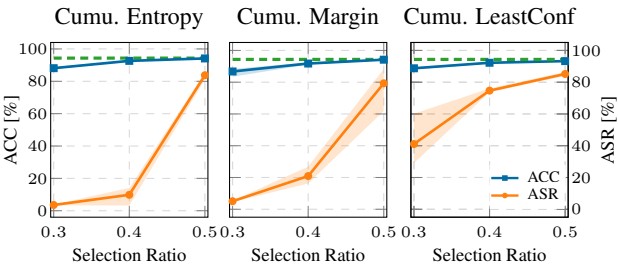

*(b)* Cumulative Uncertainty Criteria

*Figure 10.* Comparing loss-based criteria to uncertainty-based criteria with accumulation. All experiments are conducted by using ResNet18 model on CIFAR10 that is under the poisoning of WaNet attack with $\rho = 5\%$. Error bars show the value range across five random runs per coreset selection ratio.

loss-based criteria in Figure 2, these criteria above select coresets that either exhibit low stability, lack robustness of eliminating poisonous samples, leading to a high ASR after training, or even significantly degrade the natural performance after model training. Therefore, we see these criteria as unqualified methods. Hence, we focus on the uncertainty-based and loss-based methods in our main analysis.

## B. Impact of Selection Criteria for WaNet

Figure 10 presents the comparison between the naive application of loss-based criteria and uncertainty-based criteria with accumulation, under WaNet attack scenario. Unlike Blend attack case (cf. Figure 2), the EL2N method fails to effectively exclude poisonous samples during coreset selection, resulting a high ASR. This indicates that solely using loss-based criteria cannot achieve a robust elimination of poisonous samples across diverse attacks. In contrast, incorporating accumulation with uncertainty criteria yields better defensive performance, despite a relatively high ASR at a selection ratio of 0.5. Notably, at the selection ratio 0.4, using CENT achieves a natural accuracy comparable to the baseline while significantly mitigating the WaNet backdoor. This demonstrates the advantage of adopting Cumulative Entropy criterion in anti-backdoor coreset selection.

## C. Experimental Setup

In this section, we outline the experimental setups for considered attacks and defenses. Note that each experiment is conducted on one single Nvidia RTX-3090 GPU with Intel(R) Xeon(R) W-2245 CPU. Each result is conducted by 5 runs of experiment for small-scale datasets CIFAR10 and GTSRB, and 3 runs of experiment for the large-scale dataset Tiny-ImageNet.

### C.1. Considered Attacks

We generate each poisoned dataset by conforming to the anti-backdoor learning application scenario, that is, the backdoor is introduced via data poisoning only. Each attack is implemented as follows.

**BadNets (Gu et al., 2017)**: We use a colored $2 \times 2$ square pattern for dataset poisoning.

**Blend (Chen et al., 2017)**: We use the default Hello-Kitty image as the trigger pattern for poisoning CIFAR10 and GTSRB, and a random noise trigger for Tiny-ImageNet. We use a trigger opacity of $\alpha = 0.1$, as it has been proven effective to achieve an attack success rate near $100\%$.

**CLB (Turner et al., 2019)**: We poison each sample by adopting the projected gradient descent (PGD) method to generate adversarial perturbation with strength $\epsilon = {}^{16}/_{255}$ and step size $^2/_{255}$ for 30 steps.

**IAB (Nguyen & Tran, 2020)**: We train the generator on each dataset the default hyper-parameters: the backdoor probability $\rho_b = 0.1$, the cross-trigger probability $\rho_c = 0.1$, and the weighting parameter $\lambda_{div} = 1.0$ for the diversity loss term. For each dataset, we train an individual generator for constructing IAB poisoning.

**WaNet (Nguyen & Tran, 2021)**: We follow the data poisoning implementation of WaNet, i.e., first generating a warping trigger function and its accompanied noise function and, then, using both pattern functions to poison data samples. For all datasets, we following the default settings of WaNet, i.e., using the noise ratio equal $2 \times \rho$, the grid size $k = 4$, and the warping strength $s = 0.5$.

**ISSBA (Li et al., 2021a)**: For each dataset and poisoning rate, we execute the ISSBA attack by following the implementation provided by BackdoorBench (Wu et al., 2022).

**Low Frequency (LF) (Zeng et al., 2021)**: We reuse the generated triggers that are provided by BackdoorBench (Wu et al., 2022) and run the data poisoning.

**Adaptive Blend (A-Blend) (Qi et al., 2023)**: We adopt the same trigger as used in (Qi et al., 2023) and set the trigger opacity equal 0.15 for training set and 0.2 for test set. For the random trigger partitioning, we set the mask rate equal 0.5 and coverage rate equal 1.0 and use 16 masking pieces.

*Table 6.* Comparing ABCS to prior training-time defenses on GTSRB. Each result contains the averaged value and the standard deviation over five random runs. The best results across all defenses are highlighted as **boldface**. The defense failure (i.e., ASR > 50 %) is shown as **orange boldface**. We discard ASR for No-Defense as all attacks reach an ASR above 99 %.

| Attack | No-Defense | ABL | | | DBD | | | CBD | | |
|---|---|---|---|---|---|---|---|---|---|---|
| | ACC (↑) | ACC (↑) | ASR (↓) | DER (↑) | ACC (↑) | ASR (↓) | DER (↑) | ACC (↑) | ASR (↓) | DER (↑) |
| BadNets | 97.82± 0.70 | 89.30±9.59 | **53.13**± 50.23 | 69.18±28.85 | 92.36±0.42 | 0.02± 0.03 | 97.26± 0.21 | 91.80±3.11 | **80.31**±43.98 | 56.84±21.34 |
| Blend | 98.08± 0.37 | 91.29±2.07 | **94.96**± 7.21 | 48.81± 3.81 | 93.03±0.13 | **99.74**± 0.38 | 47.48± 0.07 | 86.28±8.99 | **97.42**± 2.20 | 45.02± 4.61 |
| CLB | 98.06± 0.08 | 86.64±2.95 | 7.53± 5.92 | 90.50± 1.96 | 92.98±0.11 | **0.47**± 0.16 | 97.20± 0.10 | 82.99±2.57 | 10.21±11.70 | 87.34± 6.57 |
| IAB | 98.03± 0.45 | 94.56±3.68 | 20.74±43.91 | 87.87±23.79 | 92.83±0.61 | 0.28± 0.18 | 97.23± 0.28 | 83.05±4.10 | 39.12±38.06 | 72.92±19.46 |
| WaNet | 97.44± 0.61 | 86.32±6.16 | **99.76**± 0.05 | 44.44± 3.08 | 92.64±0.31 | **0.00**± 0.00 | 95.71± 0.16 | 96.20±0.25 | **97.25**± 0.46 | 49.38± 0.12 |
| ISSBA | 98.04± 0.06 | 88.81±2.82 | **99.86**± 0.14 | 45.45± 1.38 | 92.74±0.54 | **99.46**± 0.64 | 47.62± 0.10 | 76.89±5.12 | 44.69±34.43 | 67.08±18.06 |
| LF | 97.93± 0.28 | 86.58±2.86 | **84.62**±32.42 | 52.02±15.41 | 92.32±0.48 | 27.45± 9.53 | 83.47± 4.72 | 74.79±7.44 | **77.44**±37.29 | 49.70±16.26 |
| A-Blend | 97.56± 0.29 | 92.28±3.06 | **0.22**± 0.22 | 97.19± 1.62 | 92.84±0.25 | **80.29**±18.00 | 57.46± 9.04 | 81.66±4.43 | **91.03**± 2.03 | 46.47± 1.67 |
| Average | 97.87 | 89.47 | **57.60** | 66.93 | 92.72 | 38.46 | 77.93 | 84.21 | **67.18** | 59.34 |
| WorstCase | 97.44 | 86.32 | **99.86** | 44.44 | 92.32 | **99.74** | 47.48 | 74.79 | **97.42** | 45.02 |

| Attack | No-Defense | V&B | | | Harvey | | | ABCS (Ours) | | |
|---|---|---|---|---|---|---|---|---|---|---|
| | ACC (↑) | ACC (↑) | ASR (↓) | DER (↑) | ACC (↑) | ASR (↓) | DER (↑) | ACC (↑) | ASR (↓) | DER (↑) |
| BadNets | 97.82± 0.70 | 93.35±0.73 | 0.05± 0.04 | 97.74± 0.38 | 98.19±0.20 | **0.00**± 0.00 | **100.00**± 0.00 | **98.27**±0.25 | 0.00± 0.00 | **100.00**± 0.00 |
| Blend | 98.08± 0.37 | 92.89±2.68 | **98.01**± 1.38 | 48.04± 1.15 | 98.04±0.10 | 0.05± 0.05 | **99.56**± 0.07 | **98.05**±0.32 | 0.40± 0.37 | **99.33**± 0.18 |
| CLB | 98.06± 0.08 | 91.16±0.53 | 13.00±16.88 | 90.02± 8.59 | **98.26**±0.59 | 1.04± 0.64 | **99.38**± 0.24 | 97.62±0.33 | 1.53± 0.44 | **98.98**± 0.17 |
| IAB | 98.03± 0.45 | 93.56±2.40 | **89.30**±15.36 | 53.09± 7.97 | 98.11±0.23 | 0.03± 0.06 | 99.93± 0.05 | **98.20**±0.26 | **0.01**± 0.02 | **99.94**± 0.05 |
| WaNet | 97.44± 0.61 | 94.57±1.05 | 46.04±18.25 | 73.65± 9.14 | 97.88±0.30 | 7.08± 2.86 | 94.57± 1.43 | **98.11**±0.39 | 0.01± 0.02 | **98.10**± 0.01 |
| ISSBA | 98.04± 0.06 | 91.58±1.55 | **78.74**±27.00 | 57.40±13.12 | 97.81±0.27 | **0.00**± 0.00 | 99.88± 0.12 | 97.90±0.18 | 0.01± 0.03 | **99.91**± 0.07 |
| LF | 97.93± 0.28 | 93.49±1.00 | **98.15**± 2.29 | 48.70± 0.68 | 94.99±1.84 | **99.83**± 0.20 | 48.61± 0.83 | 97.91±0.29 | 1.52± 1.69 | **99.18**± 0.81 |
| A-Blend | 97.56± 0.29 | 93.36±1.74 | **98.55**± 1.01 | 48.56± 1.22 | 94.97±2.56 | **96.66**± 1.09 | 50.23± 1.32 | **96.37**±0.98 | 7.94± 1.59 | 95.32± 1.07 |
| Average | 97.87 | 92.99 | **65.23** | 64.65 | 97.28 | 25.59 | 86.52 | **97.80** | 1.43 | **98.85** |
| WorstCase | 97.44 | 91.16 | **98.55** | 48.04 | 94.97 | **99.83** | 48.61 | **96.37** | 7.94 | **95.32** |

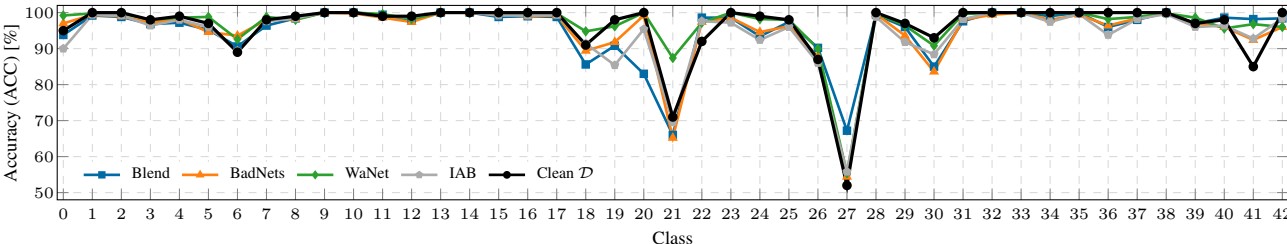

*Figure 11.* Class-wise accuracy distribution on GTSRB across different attacks, averaged over five runs. "Clean $\mathcal{D}$" refers to using the original clean dataset for the model training.

## C.2. Considered Defenses

**ABL (Li et al., 2021b)**: The ABL procedure consists of three stages: (1) training for 20 epochs on the entire poisoned dataset and isolating 1 % of the samples with the lowest loss, (2) fine-tuning the model on the remaining dataset, and (3) unlearning the isolated poisonous set for 5 epochs with a learning rate of 0.0001. The hyperparameter $\gamma$ in LGA (Local Gradient Ascent) is sensitive to different attacks. However, for practicality, we consistently use $\gamma = 0.5$ in the implementation.

**DBD (Huang et al., 2022a)**: DBD first adopts self-supervised learning for 1,000 epochs to extract benign features from the dataset. Then, it fine-tunes the fully connected layer for 10 epochs with supervised learning, splitting the dataset half-and-half, and subsequently uses semi-supervised learning to train the entire model for 200 epochs. DBD does not specify other hyperparameters for individual

backdoor attacks. Therefore, we directly follow the default settings of DBD to conduct all experiments.

**CBD (Zhang et al., 2023)**: CBD learns a model on the poisoned dataset and then trains a clean model by maximizing mutual independence from the former. Since the first model is responsible for identifying poisonous samples, the number of training epochs in the first phase is a sensitive hyperparameter for different attacks. The original implementation selects the best number from $\{3, 5, 8\}$. Accordingly, we tested different numbers of epochs and takes the result with the highest natural accuracy.

**V&B (Zhu et al., 2023b)**: The V&B defense first exploits a model with a backdoor to identify suspiciously poisoned samples and then trains a clean model on benign ones. Afterward, it adopts semi-supervised learning, as in (Huang et al., 2022a), on the split datasets to further improve the natural performance of the clean model while simultaneously

suppressing the backdoor functionality. In our evaluation, we use the default settings and hyperparameters of V&B for both the small-scale and large-scale datasets.

**Harvey (Zhao & Wressnegger, 2025)**: Harvey constructs a reference model with a backdoor through warm-up training and subsequently boosts convergence toward the backdoor by iteratively training on a poisoned subset and re-splitting the poisoned dataset. We follow the default settings described in the appendix of Harvey for all experiments.

**ABCS (Ours)**: During the coreset selection, we exclude all data augmentations to stabilize backdoor learning (Li et al., 2021b; Qiu et al., 2021). In addition to the setup in Section 5, the final training on coresets follows standard settings: models are trained for 200 epochs using the SGD optimizer with a weight decay of 0.0005, and the learning rate is scheduled via cosine annealing from 0.1 to 0.0001.

# D. Extended Evaluation

In this section, we conduct additional experiments to extend our evaluation. We first examine all defenses on GTSRB dataset (cf. Appendix D.1), and then investigate the impact of other global settings on ABCS (cf. Appendix D.2). After that, we test ABCS across different poisoning settings (cf. Appendix D.3), with various model architectures (cf. Appendix D.4) and against the all-to-all attacks (cf. Appendix D.5). Moreover, we examine the potential of ABCS for the text classification domain (cf. Appendix D.6). Finally, we compare to other defense formats that either possesses certain clean reference samples (cf. Appendix D.7) or remove backdoors after the training (cf. Appendix D.8).

## D.1. Evaluation on GTSRB

Table 6 summarizes the experiments on the GTSRB dataset. For the ResNet18 model, learning on GTSRB is easier than on CIFAR10, resulting in a natural accuracy close to 98 %. However, most prior defenses fail to consistently maintain high accuracy across all attacks. Moreover, prior defenses show a drop in ACC up to 7 % and allow at least one attack to successfully implant the backdoor into the model. In contrast, training on the coreset selected by ABCS effectively mitigates all backdoor attacks while maintaining high natural accuracy.

**Analysis of class-wise accuracy preservation.** Unlike CIFAR10 that has a uniform class-wise distribution, GTSRB is intrinsically imbalanced across its all 43 classes (Stallkamp et al., 2012). Figure 11 compares the class-wise accuracy yielded by training on coresets determined using a model trained on a fully clean dataset $\mathcal{D}$. Our method benefits from the sufficiently large size of coresets (cf. Table 7) and, thus, ensures the coverage of the dataset informativeness.

Moreover, training on the full clean dataset yields notably low accuracy at class 21 and 27. For these difficult classes, our method yields comparable or even higher accuracies, demonstrating its potential to preserve fairness.

## D.2. Impact of Other Global Settings on ABCS

**Rationale of excluding mis-predicted samples.** To illustrate the importance of automatic thresholding, we conduct additional experiments for the Blend attack across three datasets in Table 7. The selection ratio $r_{se}$ varies significantly across datasets, highlighting the necessity of an automatic threshold over a fixed one. Our approach effectively adjusts the threshold to maintain high natural accuracy, thereby improving robustness and adaptability. Moreover, our method excludes mis-prediced samples when calculating the threshold. In additon, we compare our strategy with a variant that applies the automatic threshold including mis-predicted samples as well, denoted as All Samples. The latter leads to a smaller selection ratio and consequently worse natural accuracy (especially on Tiny-ImageNet). These findings emphasize that our automatic thresholding strategy not only preserves performance but also generalizes better across different datasets.

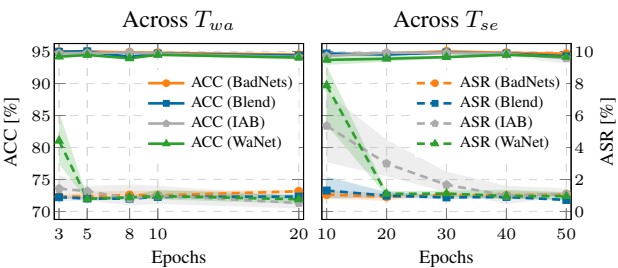

*Figure 12.* Investigating the impact of warm-up epochs $T_{wa}$ (left), selection epochs $T_{se}$ (right) on ABCS's defense for CIFAR10.

$T_{wa}$ **and** $T_{se}$**.** Based on the ablation study in Figure 12, both warm-up epochs $T_{wa}$ and selection epochs $T_{se}$ have minimal impact on natural accuracy. However, a shorter warm-up phase may fail to capture backdoor behavior, leading to less effective filtering of poisonous samples and, thus, a higher ASR in the final training. Similarly, ABCS benefits from a longer selection phase: accumulating entropy over more epochs ensures the exclusion of poisonous samples

*Table 7.* Impact of excluding mis-predicted samples during ABCS's coreset selection.

| Dataset | All Samples | | | Excl. Mis-predicted Samples | | |
|---|---|---|---|---|---|---|
| | **ACC** | **ASR** | $r_{se}$ | **ACC** | **ASR** | $r_{se}$ |
| CIFAR10 | 93.88±0.31 | 1.09±0.24 | 44.67±0.81 | **94.62**±0.36 | **0.77**±0.08 | **53.85**±0.58 |
| GTSRB | 97.39±0.42 | **0.32**±0.26 | 31.26±1.00 | **98.05**±0.32 | 0.40±0.37 | **33.13**±0.97 |
| Tiny-ImageNet | 38.53±1.02 | **0.01**±0.01 | 48.72±0.15 | **61.11**±0.07 | 0.18±0.05 | **82.30**±0.64 |

and retrain informative benign ones in the selected coreset, thereby yielding high natural accuracy and low ASR.

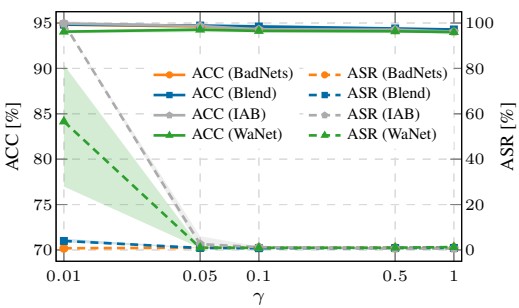

*Figure 13.* Impact of $\gamma$ on ABCS's defense for CIFAR10.

**Values of $\gamma$.** While $\gamma$ decreases, the natural accuracy (ACC) increases (cf. Figure 13). A large value of $\gamma$ ($\geq 0.1$) effectively lowers the ASR, while a smaller $\gamma$ leads to insufficient unlearning and, thus, results in ineffective separation of poisoned samples and benign samples with high CENT uncertainty, thereby yielding a high ASR in the final model.

### D.3. Evaluation Across Different Poisoning Settings

**Target classes.** We evaluate the robustness of ABCS across different target classes under variant attacks in Figure 14 (left). For each poisoned CIFAR10, we report the average performance and error bar over five random trials. Across all target classes, ABCS consistently select a coreset that enables training a model from scratch with high natural performance while keeping ASR below $2\%$.

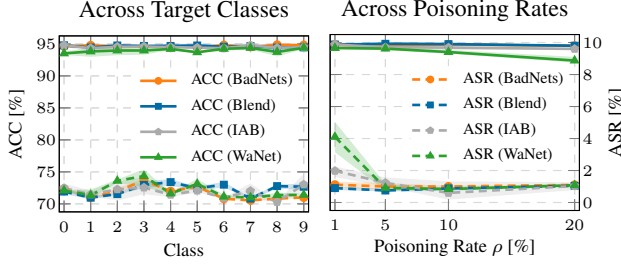

*Figure 14.* Evaluation across target classes (left) and different poisoning rates (right) on CIFAR10.

**Poisoning rates.** Figure 14 (right) shows the performance of ABCS across different poisoning rates. A higher poisoning rate $\rho$ corresponds to a stronger attack mode, while a lower $\rho$ increases the difficulty of learning backdoor. Figure 15 shows the training processes under different poisoning rates. We use $\mathcal{L}_{dis} = \mathcal{L}_{bng} - \mathcal{L}_{poi}$ to measure the loss discrepancy between benign and poisoned samples, where $\mathcal{L}_{bng}$ and $\mathcal{L}_{poi}$ are

the mean loss value of benign and poisoned samples, respectively. A lower poisoning rate will slow down the convergence speed to the backdoor, thus, making the loss discrepancy become smaller in the initial epochs. For $\rho \geq 5\%$, ABCS consistently selects a coreset that yields a model with high natural accuracy and minimal ASR, indicating effective exclusion of poisonous samples. At a lower poisoning rate ($\rho = 1\%$), the increased difficulty of

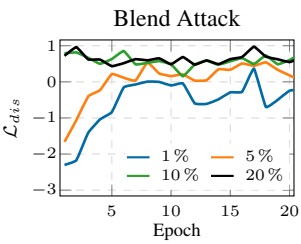

*Figure 15.* Impact of varying $\rho$.

learning backdoor slightly raises the risk of including several poisoned samples in the coreset. Despite this, ABCS continues to perform robustly with anti-backdoor coreset selection, ensuring a strong mitigation of backdooring attacks (i.e., ASR $< 5\%$) while maintaining high natural accuracy.

### D.4. Evaluation with Different Model Architectures

We evaluate ABCS across three additional model architectures, i.e.,, VGG11 (Simonyan & Zisserman, 2015), MobileNetV2 (Sandler et al., 2018) and DenseNet121 (Huang et al., 2017), under all considered attacks. As shown in Table 8, training on the coresets selected by ABCS consistently yields natural accuracy comparable to that of training on a clean dataset. Simultaneously, the low ASR demonstrates a strong backdoor mitigation, highlighting effective elimination of poisonous samples during coreset selection.

### D.5. Evaluation on All-to-All Attacks

We set up all-to-all attacks with the same settings as (Li et al., 2022), i.e., taking the next neighbor class as the backdoor target of each poisoned sample. Additionally, we extend the setting to using the Hello-Kitty trigger of Blend attack. As shown in Table 9, our method remains effective against both two all-to-all attacks, showing a high robustness our method.

*Table 9.* ABCS's defense against all-to-all attacks on CIFAR10.

| All-to-All | ACC | ASR | DER |
|---|---|---|---|
| BadNets | 94.36±0.30 | 0.56±0.05 | 94.19±0.46 |
| Blend | 94.00±0.16 | 1.86±0.86 | 93.34±0.40 |

### D.6. Evaluation on Text Classification Task

Following Cheng et al. (2025), we adopt the standard data-poisoning setting (Kurita et al., 2020), where the trigger is a rare word "cf" inserted randomly into each poisoned text sample, and use the target label 0. Additionally, we reduce the poisoning rate to $5\%$. We implement this attack on two versions of the Stanford Sentiment Treebank (SST) dataset: SST-2 (binary sentiment classification) and SST-5 (five sentiment classes) using pre-trained BERT-base (uncased) and

*Table 8.* Evaluation of ABCS across model architectures on CIFAR10. For each model, ACC from training on the clean dataset is shown next to the model name. In the table, ACC results (in [%]) correspond to training on selected coresets from each poisoned dataset. For backdoor mitigation, ASR (in [%]) is reported for training on the poisoned dataset and on the selected coreset derived by ABCS (shown to the left and right of " / ", respectively).

| Attack | VGG11 (91.85 %) | | MobileNetV2 (93.15 %) | | DenseNet121 (95.16 %) | |
|---|---|---|---|---|---|---|
| | ACC (↑) | ASR (↓) | ACC (↑) | ASR (↓) | ACC (↑) | ASR (↓) |
| BadNets | $91.76_{\pm0.08}$ | $99.98_{\pm0.11}$ / $1.32_{\pm0.22}$ | $92.93_{\pm0.07}$ | $99.96_{\pm0.06}$ / $0.99_{\pm0.13}$ | $94.97_{\pm0.26}$ | $100.00_{\pm0.00}$ / $0.83_{\pm0.06}$ |
| Blend | $90.88_{\pm0.85}$ | $99.12_{\pm0.31}$ / $2.00_{\pm0.23}$ | $92.84_{\pm0.02}$ | $99.23_{\pm0.25}$ / $1.14_{\pm0.18}$ | $94.65_{\pm0.15}$ | $99.25_{\pm0.32}$ / $0.90_{\pm0.45}$ |
| CLB | $91.38_{\pm0.11}$ | $99.62_{\pm0.25}$ / $2.90_{\pm0.29}$ | $92.83_{\pm0.16}$ | $99.87_{\pm0.13}$ / $1.18_{\pm0.16}$ | $94.60_{\pm0.02}$ | $99.48_{\pm0.15}$ / $0.90_{\pm0.15}$ |
| IAB | $91.48_{\pm0.35}$ | $99.84_{\pm0.35}$ / $1.62_{\pm0.39}$ | $92.87_{\pm0.27}$ | $99.94_{\pm0.04}$ / $2.37_{\pm0.38}$ | $94.98_{\pm0.21}$ | $99.69_{\pm0.28}$ / $1.25_{\pm0.33}$ |
| WaNet | $89.38_{\pm1.05}$ | $96.43_{\pm0.84}$ / $2.65_{\pm0.64}$ | $92.44_{\pm0.08}$ | $97.25_{\pm0.68}$ / $1.93_{\pm0.31}$ | $94.86_{\pm0.09}$ | $98.35_{\pm0.32}$ / $0.82_{\pm0.08}$ |
| ISSBA | $91.16_{\pm0.29}$ | $99.07_{\pm0.29}$ / $1.83_{\pm0.26}$ | $92.50_{\pm0.26}$ | $99.21_{\pm0.32}$ / $1.66_{\pm0.24}$ | $94.71_{\pm0.22}$ | $99.06_{\pm0.41}$ / $0.80_{\pm0.37}$ |
| LF | $91.22_{\pm0.38}$ | $98.14_{\pm0.74}$ / $7.22_{\pm1.86}$ | $92.48_{\pm0.10}$ | $98.31_{\pm0.57}$ / $6.66_{\pm1.97}$ | $94.52_{\pm0.18}$ | $98.36_{\pm0.89}$ / $5.83_{\pm1.61}$ |
| A-Blend | $90.82_{\pm0.29}$ | $99.04_{\pm0.54}$ / $9.02_{\pm0.79}$ | $91.93_{\pm0.47}$ | $99.52_{\pm0.22}$ / $7.82_{\pm1.07}$ | $94.47_{\pm0.11}$ | $99.42_{\pm0.51}$ / $7.32_{\pm0.95}$ |

BERT-large (uncased) models (Devlin et al., 2019) as basis. To stabilize the fine-tuning process during coreset selection on the pre-trained BERT model, we use the AdamW optimizer with a learning rate of $2e-6$. For training the final coreset, the learning rate is increased to $2e-5$. Due to the faster convergence during fine-tuning the pre-trained models, we set $T_{warm} = 5$ epochs for warm-up and $T_{se} = 10$ epochs for coreset selection. The resulting coreset is then used to fine-tune the original pre-trained BERT model for 10 epochs. As shown in Table 10, our method adaptively determines different selection ratios $r_{se}$ for SST-2 and SST-5. Despite a slight drop in accuracy for SST-5, the results show that our method remains robust against data poisoning backdoors in text classification tasks.

*Table 10.* Effectiveness of ABCS on text classification task.

| BERT Model | Dataset | No-Defense | | ABCS | | | |
|---|---|---|---|---|---|---|---|
| | | ACC | ASR | ACC | ASR | $\rho_{bng}$ | $r_{se}$ |
| Base | SST-2 | $92.33_{\pm0.31}$ | $99.97_{\pm0.04}$ | $91.63_{\pm0.52}$ | $6.89_{\pm0.97}$ | $0.01_{\pm0.02}$ | $34.35_{\pm0.40}$ |
| | SST-5 | $49.83_{\pm0.37}$ | $99.45_{\pm0.33}$ | $48.20_{\pm0.82}$ | $3.53_{\pm0.58}$ | $0.00_{\pm0.00}$ | $57.21_{\pm1.97}$ |
| Large | SST-2 | $93.39_{\pm0.98}$ | $99.92_{\pm0.12}$ | $92.50_{\pm0.87}$ | $6.23_{\pm0.91}$ | $0.01_{\pm0.01}$ | $35.44_{\pm0.30}$ |
| | SST-5 | $50.80_{\pm0.87}$ | $99.75_{\pm0.09}$ | $48.53_{\pm0.73}$ | $3.40_{\pm0.32}$ | $0.00_{\pm0.00}$ | $56.72_{\pm0.51}$ |

In addition, we compare with SEEP (He et al., 2024), a typical defense against data-poisoning backdoors in text classification. Due to the lack of publicly available code, we follow SEEP's reported poisoning settings and evaluate on the SST-2 dataset under BadNet (Kurita et al., 2020) and Syntactic (Qi et al., 2021) attacks. ABCS preserves high ACC while effectively eliminating poisoned samples, achieving defensive performance comparable to SEEP (cf. Table 11).

*Table 11.* Comparing ABCS to SEEP.

| Attack | SEEP | ABCS (ours) |
|---|---|---|
| | ACC / ASR | ACC / ASR |
| BadNet | **92.6** / 7.4 | 92.3 / **4.9** |
| Syntactic | 91.5 / **10.0** | **91.7** / 12.7 |

## D.7. Comparison to Defense with Clean Reference Data

While assuming the absence of any reference dataset is most practical, this setting imposes a strong restriction. Many post-training defenses thus permit the access to a small portion of clean data. A representative method is ASD, the Adaptively Splitting Defense (Gao et al., 2023), which picks 10 clean samples per class uniformly of random for model initialization. With the prior knowledge of benign samples, the subsequent adaptive dataset splitting can more effectively isolate poisoned samples due to their higher prediction losses than benign ones.

*Table 12.* Comparing ABCS to Adaptively Splitting Defense (ASD) on CIFAR10. Each result contains the averaged value and the standard deviation over five random runs. The best results are highlighted as **boldface**. ASR for No-Defense is discarded as all attacks reaches a ASR over 99 %.

| Attack | ASD | | | ABCS (Ours) | | |
|---|---|---|---|---|---|---|
| | ACC (↑) | ASR (↓) | DER (↑) | ACC (↑) | ASR (↓) | DER (↑) |
| BadNets | $93.56_{\pm0.39}$ | $1.39_{\pm0.52}$ | $98.82_{\pm0.29}$ | $\mathbf{94.38}_{\pm0.23}$ | $\mathbf{1.00}_{\pm0.17}$ | $\mathbf{99.42}_{\pm0.19}$ |
| Blend | $93.31_{\pm0.65}$ | $5.50_{\pm1.78}$ | $95.30_{\pm0.79}$ | $\mathbf{94.62}_{\pm0.36}$ | $\mathbf{0.77}_{\pm0.08}$ | $\mathbf{98.22}_{\pm0.07}$ |
| CLB | $92.92_{\pm0.64}$ | $1.30_{\pm0.90}$ | $98.58_{\pm0.30}$ | $\mathbf{94.51}_{\pm0.23}$ | $\mathbf{0.88}_{\pm0.10}$ | $\mathbf{99.54}_{\pm0.06}$ |
| IAB | $93.42_{\pm0.45}$ | $2.25_{\pm1.03}$ | $98.23_{\pm0.48}$ | $\mathbf{94.30}_{\pm0.16}$ | $\mathbf{1.22}_{\pm0.37}$ | $\mathbf{99.19}_{\pm0.19}$ |
| WaNet | $92.63_{\pm0.41}$ | $17.29_{\pm21.24}$ | $88.67_{\pm10.78}$ | $\mathbf{94.13}_{\pm0.22}$ | $\mathbf{0.89}_{\pm0.16}$ | $\mathbf{97.53}_{\pm0.13}$ |
| ISSBA | $92.80_{\pm0.35}$ | $2.26_{\pm0.67}$ | $98.01_{\pm0.36}$ | $\mathbf{94.66}_{\pm0.41}$ | $\mathbf{1.22}_{\pm0.09}$ | $\mathbf{99.33}_{\pm0.15}$ |
| LF | $93.72_{\pm0.23}$ | $5.66_{\pm3.67}$ | $95.85_{\pm1.78}$ | $\mathbf{94.06}_{\pm0.30}$ | $\mathbf{3.04}_{\pm0.55}$ | $\mathbf{97.33}_{\pm0.17}$ |
| A-Blend | $93.24_{\pm0.38}$ | $34.84_{\pm6.99}$ | $80.18_{\pm3.44}$ | $\mathbf{94.37}_{\pm0.27}$ | $\mathbf{5.71}_{\pm1.59}$ | $\mathbf{95.12}_{\pm0.77}$ |
| Average | 93.20 | 8.81 | 94.20 | **94.38** | **1.84** | **98.21** |
| WorstCase | 92.63 | 34.84 | 80.18 | **94.06** | **5.71** | **95.12** |

In Table 12, we evaluate ABCS's performance on CIFAR10 against ASD. Unlike other defenses that often fail (cf. Table 1a), ASD shows a higher robustness across diverse attacks. Nonetheless, its performance remains limited against A-Blend and WaNet, and it struggles to preserve natural accuracy. In contrast, ABCS consistently produces an informative, clean coreset that supports training with minimal ASR while maintaining accuracy comparable to the original model. Although the reference data facilitates more reliable dataset splitting, it is intrinsically less effective in a training progress with dynamics. ABCS circumvents this limitation by extracting a backdoor-free coreset, enabling a

more reliable and robust training-time backdoor defense.

### D.8. Comparison to Post-train Defenses

In addition, we compare ABCS to post-training defenses, ANP (Wu & Wang, 2021) and SAU (Wei et al., 2023), which aim to remove backdoors from models pre-trained on poisoned datasets. Both methods are implemented using the BackdoorBench (Wu et al., 2022), with ResNet18 on CIFAR10. Results are reported in Table 13. Although both defenses leverage some clean reference samples, they achieve effective backdoor removal at the cost of reduced natural accuracy. In contrast, ABCS preserves high natural accuracy while still strongly suppressing backdoors, demonstrating its effectiveness in selecting a backdoor-free coreset.

*Table 13.* Comparing ABCS to post-train defenses.

| Attack | ANP | SAU | ABCS |
|---|---|---|---|
| | ACC / ASR | ACC / ASR | ACC / ASR |
| BadNets | 92.08 / 1.27 | 92.97 / 2.72 | **94.38** / **1.00** |
| Blend | 88.55 / 4.40 | 91.64 / 1.20 | **94.62** / **0.77** |
| IAB | 93.08 / 2.08 | **94.91** / **0.45** | 94.30 / 1.22 |
| WaNet | 92.48 / **0.60** | 93.11 / 1.81 | **94.13** / 0.89 |

## E. Robustness Against Adaptive Attacks

Anti-backdoor learning allows a full access to the dataset without any control of the training process. Under this setting, we evaluate ABCS against five different adaptive attacks that attempt to circumvent our defense by either increasing the predictive uncertainty of poisoned samples (cf. Appendix E.1 and E.2), intentionally restricting the training dataset utility (cf. Appendix E.3), or enforcing a larger learning difficulty for poisoned samples (cf. Appendix E.4, cf. Appendix E.5).

### E.1. Poisoning Samples Using CENT Ranking

We first assume that the adversary understands the coreset selection strategy of using the CENT criterion. Given that, we conduct two possible adversaries that utilizes the ranking of CENT for the dataset poisoning.

**One-shot poisoning of informative samples.** Intuitively, the adversary can first rank all clean samples according to the order of Cumulative Entropy and only poisons $\rho = 5\%$ samples with the highest uncertainty. This way, poisoned samples contain hard-to-learn features, thereby implicitly increasing their prediction uncertainty during model training. Table 14 summarizes the experiments of ABCS on CIFAR10 with a ResNet18 model against diverse attacks using the adaptive method, where we exclude CLB attack due to its clean-label poisoning constraint. Compared to Table 1, poisoning informative samples increases the difficulty of learning backdoors, thereby reducing the attack success rate

after naive training but also complicating the anti-backdoor coreset selection. As a result, ABCS yields a slightly higher ASR under adaptive attacks. Nevertheless, ABCS effectively selects coresets from the entire dataset, maintaining high natural accuracy while significantly reducing ASR.

*Table 14.* ABCS against adaptive attacks using CENT ranking. For both ACC and ASR, we show the results of No-Defense and ABCS on the left and right of " / ", respectively

| Attack | ACC ($\uparrow$) | ASR ($\downarrow$) |
|---|---|---|
| BadNets | $94.22_{\pm0.05}$ / $93.98_{\pm0.20}$ | $99.94_{\pm0.04}$ / $1.31_{\pm0.22}$ |
| Blend | $94.11_{\pm0.11}$ / $94.01_{\pm0.11}$ | $92.19_{\pm1.09}$ / $1.80_{\pm0.92}$ |
| IAB | $94.42_{\pm0.11}$ / $93.98_{\pm0.22}$ | $99.55_{\pm0.16}$ / $3.58_{\pm3.71}$ |
| WaNet | $93.87_{\pm0.18}$ / $92.15_{\pm0.28}$ | $79.06_{\pm2.47}$ / $6.44_{\pm2.80}$ |
| ISSBA | $94.13_{\pm0.09}$ / $94.06_{\pm0.09}$ | $98.01_{\pm0.35}$ / $1.27_{\pm0.24}$ |
| LF | $94.37_{\pm0.09}$ / $93.29_{\pm0.24}$ | $92.60_{\pm1.48}$ / $8.20_{\pm1.07}$ |
| A-Blend | $94.39_{\pm0.15}$ / $93.99_{\pm0.41}$ | $99.98_{\pm0.02}$ / $11.69_{\pm2.19}$ |

---

**Algorithm 1** Greedy Search of Poisoned Samples

---

**Input:** Clean dataset $\mathcal{D}$, a ResNet18 as $f_\theta$, the number of iteration $T$ and target poisoning rate $\rho$
**Output:** A poisoned dataset $\tilde{\mathcal{D}}$

**Initialisation:**
Train $f_\theta$ for 40 epochs on $\mathcal{D}$ and poison 50% samples with highest CENT as $\tilde{\mathcal{D}}^{(0)}$

Compute step size $n = \text{clip}\left(\dfrac{(0.5 - \rho) \cdot |\mathcal{D}|}{T}\right)$

**for** $t \in [1, \ldots, T]$ **do**
    Step 1: Train $f_\theta$ from scratch for 40 epochs
    Step 2: Calculate CENT for all poisoned samples
    Step 3: Sort poisoned samples and recover $n$ samples with the lowest CENT to benign
    Step 4: Obtain poisoned $\tilde{\mathcal{D}}^{(t)}$ with $(0.5 \times |\mathcal{D}| - n)$ poisoned samples
**end for**

---

**Few-shot poisoning of informative samples.** We consider another adversary that increases the CENT of poisoned samples by using a greedy search strategy, which begins with a high poisoning ratio ($\rho = 50\%$) and iteratively recovers a portion of poisoned samples with low CENT values back to their originally benign ones, finally reaching the target poisoning ratio. The full procedure is described in Algorithm 1. For typical attacks such as Blend and WaNet, we run for 5 and 10 iterations (cf. Table 15), where the latter is observed to converge. Across both settings, exploring poisoned samples with high CENT scores does not realistically improve the uncertainty of poisoned samples close to those of informative and benign samples. Using ABCS remains effective of extracting clean and informative coreset.

### E.2. Poisoning a Single Source Class

Given the threat model, an adversary can poison the dataset arbitrarily. One might be class-selective, where all poisoned samples originate from a single class, leading to a

*Table 15.* ABCS against few-shot poisoning with CENT ranking. For both ACC and ASR, we show the results of No-Defense and ABCS on the left and right of " / ", respectively

|  | Attack | ACC (↑) | ASR (↓) |
|---|---|---|---|
| $T = 5$ | Blend | $94.89_{\pm 0.24}$ / $94.85_{\pm 0.16}$ | $99.94_{\pm 0.03}$ / $0.94_{\pm 0.14}$ |
|  | WaNet | $94.60_{\pm 0.22}$ / $94.47_{\pm 0.24}$ | $91.18_{\pm 2.61}$ / $3.16_{\pm 2.68}$ |
| $T = 10$ | Blend | $94.85_{\pm 0.32}$ / $94.43_{\pm 0.27}$ | $99.96_{\pm 0.02}$ / $1.46_{\pm 0.48}$ |
|  | WaNet | $94.28_{\pm 0.14}$ / $94.55_{\pm 0.39}$ | $96.25_{\pm 2.45}$ / $2.35_{\pm 1.33}$ |

distribution shift between clean and poisoned data. Such imbalanced poisoning could, in principle, increase prediction uncertainty or learning stochasticity associated with the backdoor trigger, potentially challenging the defensive robustness of ABCS.

To directly evaluate this scenario, we treat each non-target class as the sole poisoning source and reduce the overall poisoning rate $\rho$ to 1 % to preserve the majority of clean samples in the chosen source class. As shown in Figure 16, we conduct experiments across all source classes on CIFAR10. Our results show that natural accuracy remains stable regardless of the poisoning source. ABCS consistently produces a coreset that ensures a high post-training accuracy across all attack types. More importantly, our method remains robust even under this highly non-uniform poisoning setting and achieves ASR below 5 % on average. These results demonstrate that ABCS is resilient to class-selective and distribution-shifted poisoning.

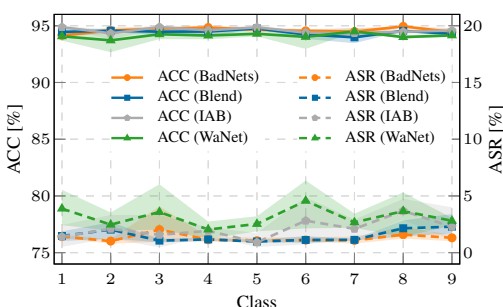

*Figure 16.* Poisoning Single Source Class.

### E.3. Constructing Low-Redundancy Training Datasets

Given adaptive adversaries aware of the coreset selection mechanism via CENT criterion, they may first extract a coreset with a selection ratio of 50 % and subsequently poison it before releasing it as the final training dataset. Due to the high informativeness of this coreset, applying anti-backdoor coreset selection becomes challenging due to the inherent trade-off between security and utility in the poisoned data. In Table 16, we additionally evaluate ABCS under such coreset-level poisoning attacks. ABCS remains effective in identifying backdoor-free coresets, yielding significantly lower ASR than No-Defense, although maintaining suffi-

cient data coverage becomes more difficult. Compared to the No-Defense baseline, ABCS selects a slightly smaller coreset, leading to a modest drop in ACC. Importantly, it automatically converges to a relatively large selection ratio $r_{se}$ (with respect to the original training coreset), thereby retaining as many samples as possible to preserve accuracy. This behavior is consistent with observations on the large-scale dataset Tiny-ImageNet (cf. Table 1b), highlighting the adaptability of ABCS in low-redundancy settings.

*Table 16.* Defending backdoors in CIFAR10 coresets ($r_{se} = 50$ %).

| Attack | Defense | ACC (↑) | ASR (↓) | $\rho_{bng}$ | $r_{se}$ |
|---|---|---|---|---|---|
| BadNets | No-Defense | $94.68_{\pm 0.24}$ | $98.44_{\pm 0.06}$ | / | / |
|  | ABCS | $93.52_{\pm 0.46}$ | $1.20_{\pm 0.38}$ | $0.00_{\pm 0.00}$ | $77.04_{\pm 0.62}$ |
| Blend | No-Defense | $94.72_{\pm 0.32}$ | $89.84_{\pm 1.35}$ | / | / |
|  | ABCS | $93.14_{\pm 0.39}$ | $1.34_{\pm 0.87}$ | $0.02_{\pm 0.02}$ | $76.20_{\pm 0.74}$ |
| IAB | No-Defense | $94.62_{\pm 0.34}$ | $96.87_{\pm 1.40}$ | / | / |
|  | ABCS | $92.85_{\pm 0.31}$ | $1.07_{\pm 0.82}$ | $0.02_{\pm 0.01}$ | $76.20_{\pm 0.56}$ |
| WaNet | No-Defense | $93.92_{\pm 0.74}$ | $71.86_{\pm 4.25}$ | / | / |
|  | ABCS | $92.02_{\pm 1.12}$ | $4.40_{\pm 2.25}$ | $1.38_{\pm 0.45}$ | $75.48_{\pm 1.26}$ |

### E.4. Trigger Randomization Attack

With full access to the training dataset, an adversary can randomize the assignment of trigger patterns from different attacks across training samples. This reduces the poisoning rate of individual attacks and simultaneously increases the uncertainty of poisoned samples. We select candidate triggers of BadNets, Blend, IAB and WaNet. During poisoning, we randomly assign one trigger to each selected sample. We use $\Delta H = H_{bng} - H_{poi}$ to show the gap between the mean uncertainty values of benign and poisoned samples, where the uncertainty value is normalized to the range $0 - 1$ at each epoch. Compared to Blend attack, trigger randomization makes poisoned samples converge more slowly,

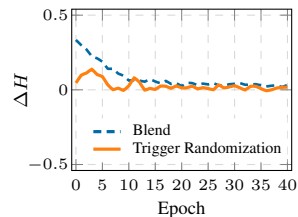

*Figure 17.* Comparing learning procedures of Blend and trigger randomization attacks.

therefore showing a smaller uncertainty distance to benign samples (cf. Figure 17). Nevertheless, our defense remains highly robust against the trigger randomization backdoor (cf. Table 17), showing strong adaptability.

*Table 17.* Defending trigger randomization attack for CIFAR10.

| Defense | ACC (↑) | ASR (↓) | | | |
|---|---|---|---|---|---|
|  |  | BadNets | Blend | IAB | WaNet |
| No-Defense | $94.63_{\pm 0.08}$ | $99.92_{\pm 0.04}$ | $91.30_{\pm 0.96}$ | $99.66_{\pm 0.04}$ | $75.60_{\pm 1.37}$ |
| ABCS (Ours) | $94.10_{\pm 0.12}$ | $1.40_{\pm 0.34}$ | $0.47_{\pm 0.08}$ | $0.91_{\pm 0.38}$ | $2.06_{\pm 0.27}$ |

*Table 18.* Investigating robustness of ABCS against adaptive attack using label randomization.

| Attack | Defense | 50 % | | 30 % | | 10 % | |
|---|---|---|---|---|---|---|---|
| | | ACC (↑) | ASR (↓) | ACC (↑) | ASR (↓) | ACC (↑) | ASR (↓) |
| BadNets | No-Defense | $95.09_{\pm0.14}$ | $73.24_{\pm2.63}$ | $94.78_{\pm0.07}$ | $91.95_{\pm0.82}$ | $94.64_{\pm0.12}$ | $99.26_{\pm1.43}$ |
| | ABCS (ours) | $93.80_{\pm0.26}$ | $16.63_{\pm2.90}$ | $93.94_{\pm0.26}$ | $6.63_{\pm2.90}$ | $94.25_{\pm0.44}$ | $3.36_{\pm0.93}$ |
| Blend | No-Defense | $94.61_{\pm0.20}$ | $64.68_{\pm4.81}$ | $94.88_{\pm0.15}$ | $85.06_{\pm0.43}$ | $94.72_{\pm0.28}$ | $94.98_{\pm0.34}$ |
| | ABCS (ours) | $93.22_{\pm0.35}$ | $9.74_{\pm6.99}$ | $93.13_{\pm0.24}$ | $3.92_{\pm2.35}$ | $94.13_{\pm0.30}$ | $2.36_{\pm0.60}$ |
| IAB | No-Defense | $94.84_{\pm0.06}$ | $71.25_{\pm1.72}$ | $95.07_{\pm0.33}$ | $89.46_{\pm1.60}$ | $94.82_{\pm0.25}$ | $98.76_{\pm0.85}$ |
| | ABCS (ours) | $93.30_{\pm0.50}$ | $30.03_{\pm4.80}$ | $93.51_{\pm0.48}$ | $9.82_{\pm1.35}$ | $94.73_{\pm0.14}$ | $5.74_{\pm0.56}$ |
| WaNet | No-Defense | $94.43_{\pm0.19}$ | $78.56_{\pm0.25}$ | $94.46_{\pm0.21}$ | $80.85_{\pm2.15}$ | $94.43_{\pm0.46}$ | $90.47_{\pm1.61}$ |
| | ABCS (ours) | $93.21_{\pm0.49}$ | $8.69_{\pm1.71}$ | $93.75_{\pm0.26}$ | $3.38_{\pm0.74}$ | $93.97_{\pm0.22}$ | $3.29_{\pm0.28}$ |
| ISSBA | No-Defense | $94.51_{\pm0.06}$ | $65.94_{\pm2.86}$ | $94.37_{\pm0.11}$ | $89.43_{\pm2.96}$ | $94.49_{\pm0.21}$ | $94.08_{\pm1.87}$ |
| | ABCS (ours) | $94.16_{\pm0.54}$ | $18.65_{\pm1.76}$ | $93.91_{\pm0.15}$ | $3.72_{\pm0.56}$ | $94.34_{\pm0.33}$ | $1.37_{\pm0.13}$ |
| LF | No-Defense | $94.87_{\pm0.01}$ | $71.36_{\pm3.62}$ | $94.89_{\pm0.05}$ | $88.00_{\pm1.72}$ | $94.77_{\pm0.15}$ | $97.00_{\pm0.24}$ |
| | ABCS (ours) | $93.67_{\pm0.53}$ | $23.99_{\pm4.97}$ | $93.53_{\pm0.28}$ | $11.45_{\pm3.64}$ | $94.55_{\pm0.58}$ | $8.93_{\pm2.53}$ |
| A-Blend | No-Defense | $94.32_{\pm0.05}$ | $33.14_{\pm3.09}$ | $94.33_{\pm0.28}$ | $34.35_{\pm2.42}$ | $94.80_{\pm0.13}$ | $65.61_{\pm4.74}$ |
| | ABCS (ours) | $93.12_{\pm0.33}$ | $15.69_{\pm2.53}$ | $93.14_{\pm0.11}$ | $15.89_{\pm4.86}$ | $93.62_{\pm0.34}$ | $13.94_{\pm3.87}$ |

### E.5. Label Randomization for Poisoned Samples

Training on datasets with randomly assigned label has been shown to increase learning difficulty (Natarajan et al., 2013), making the optimization process more stochastic due to the label randomization and thus inducing higher predictive uncertainty (Huang et al., 2022b; Köhler et al., 2019). Assuming an adversary knows the design of ABCS, an adaptive strategy would aim to increase the uncertainty of poisoned samples and slow their convergence, thereby hindering the defense. To evaluate this scenario, we assess our method under poisoning schemes that introduce label randomization to varying fractions of poisoned data. For example, with a poisoning ratio $\rho = 5\%$, a random-labeling ratio of $30\%$ means that $30\%$ of the generated poisoned samples receive randomized labels, while the remaining $3.5\%$ of poisoned samples maintain the backdoor target label.

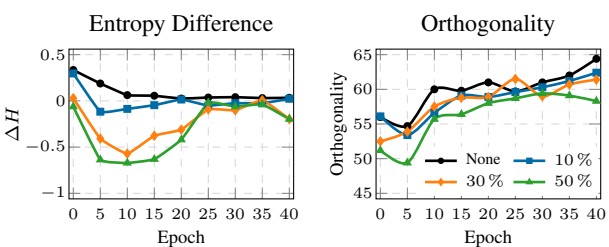

*Figure 18.* Impact of randomized label on Blend attack.

We first use the Blend attack to visualize how random labeling affects model learning. Since coreset selection relies on the uncertainty gap between benign and poisoned data, we measure the intermediate entropy difference as $\Delta H = H_{bng} - H_{poi}$, where $H_{bng}$ and $H_{poi}$ denote the mean 0–1 normalized entropy at each epoch of benign and poisoned samples. A value $\Delta H > 0$ suggests faster learning of poisoned samples due to lower entropy, whereas $\Delta H \leq 0$ indicates increased difficulty for our defense, as

noise raises the prediction uncertainty of poisoned data. As shown in Figure 18, increasing the random-labeling ratio ($10\%, 30\%, 50\%$) slows down the convergence of poisoned samples, often yielding $\Delta H < 0$ within the first 30 epochs. Furthermore, we investigate the orthogonality (Zhang et al., 2024), that is, the angle between gradient directions of benign and poisoned samples. In the loss landscape, poisoned samples exhibit gradients highly orthogonal to clean samples, enabling a more effective learning of the backdoor. With increasing the random-labeling ratio, the orthogonality decreases, showing again the effect of reducing the dissimilarity of benign and poisonous samples in the learning behavior. In Table 18, we report results across all attacks and show a trade-off: while larger random-labeling ratios increase uncertainty for poisoned samples, they simultaneously reduce the ASR. Under moderate noise (e.g., $10\%$), ABCS consistently reduces ASR, demonstrating a strong robustness even against a highly knowledgeable adversary.

## F. Empirical Analysis of Coreset Coverage

In this section, we further examine how well variant-selected coresets cover the full clean training dataset $\mathcal{D}$ under both clean and poisoned conditions. As a clean baseline with strong natural performance, we first train a ResNet18 on the complete clean training set of CIFAR10. A coreset that exhibits high visual and quantitative similarity to the full clean dataset $\mathcal{D}$ reflects a correspondingly high level of information coverage.

**Latent space visualization.** We extract the feature-space representations from the penultimate layer of ResNet18, i.e., the activation vectors immediately preceding the fully connected layer, and visualize their 2-D distributions using t-SNE (van der Maaten & Hinton, 2008). Compared with the full clean dataset $\mathcal{D}$ (cf. Figure 19a, top), selected coresets

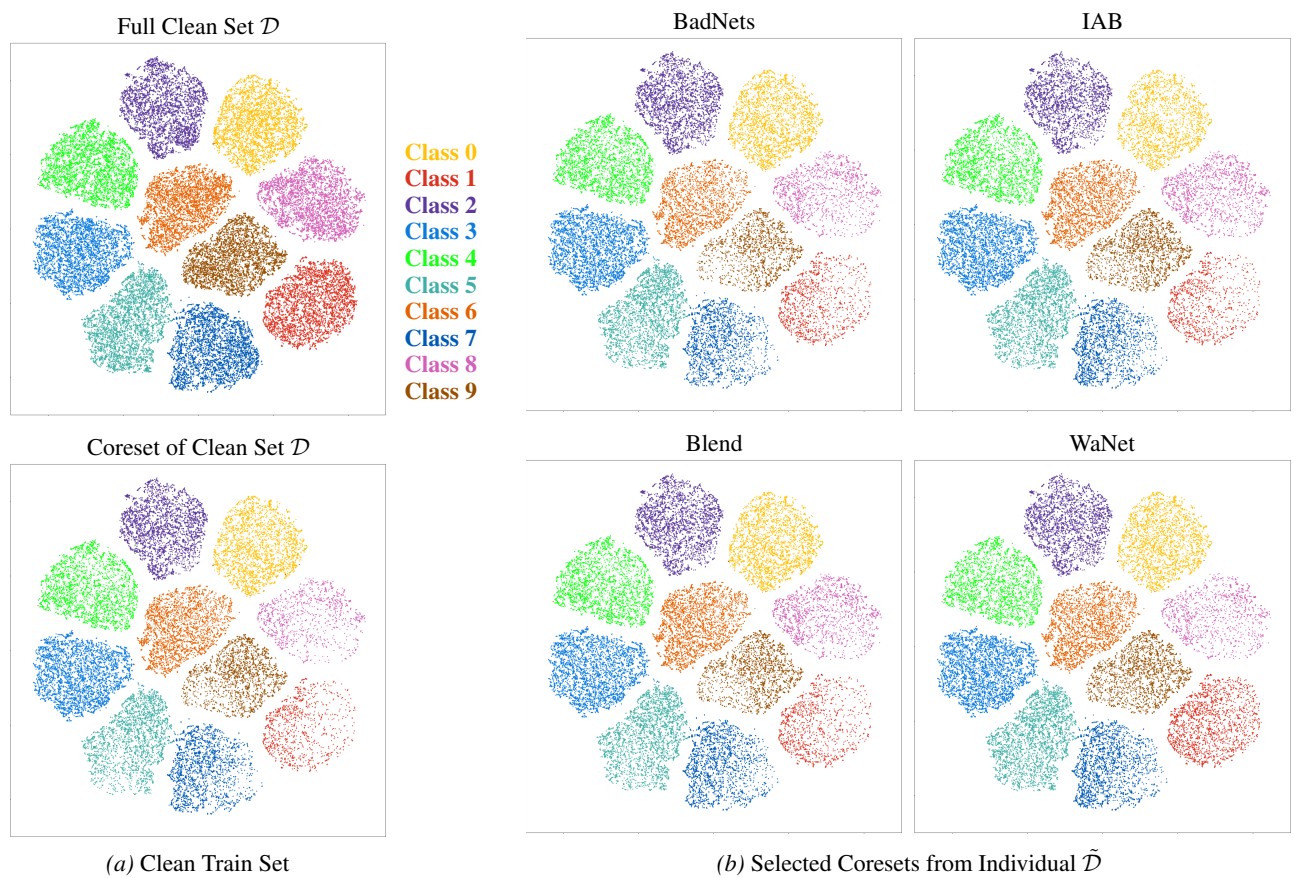

*(a)* Clean Train Set        *(b)* Selected Coresets from Individual $\tilde{\mathcal{D}}$

*Figure 19.* T-SNE visualization of data distribution of full clean training set $\mathcal{D}$ and its coreset (Figure 19a), and other coresets selected from individual poisoned training sets $\tilde{\mathcal{D}}$ (Figure 19b).

from various poisoned datasets shown in Figure 19b exhibit strong similarity in their global geometric structure. Aside from the easy classes 1 and 8 (cf. Figure 6), each remaining class preserves a cluster profile closely resembling that of the full dataset. This property consistently appears both in coresets derived from poisoned datasets and in those taken from the full clean dataset (cf. Figure 19a, bottom).

**Quantitative evaluation.** Furthermore, we conduct a quantitative evaluation along two dimensions: local geometric coverage and global distributional similarity. For the former aspect, we measure the $95^{th}$-percentile Nearest-Neighbor distance (p95-NN) (Clark & Evans, 1954), using cosine distance as the metric. For the latter aspect, we compute the Maximum Mean Discrepancy (MMD) (Borgwardt et al., 2006; Goodfellow et al., 2015), where inputs are first $l_2$-normalized and an RBF kernel is applied with bandwidth chosen via the median heuristic. Both metrics are implemented using the Scikit-learn toolkit with the default settings (Pedregosa et al., 2011). Notably, the p95-NN metric yields values in $[0, 2]$, where values close to $0$ indicate

strong local geometric coverage relative to the full clean dataset. MMD, in turn, is non-negative and attains $0$ only for identical distributions and, thus, the larger the value, the lower the global similarity.

Table 19 summarizes the similarity of the selected coresets with respect to the full clean dataset. Across all poisoning attacks, ABCS yields coresets with p95-NN and MMD under $0.015$, indicating that coreset samples lie very close to full-dataset samples and that the coreset distribution closely matches the full latent feature space distribution. These small values demonstrate both local geometric coverage and high distributional similarity of the coreset relative to the full clean dataset. Although some variance appears across classes, most of them have p95-NN and MMD values under $0.02$. Note that class-1 (that is comparably easy to learn) tolerates higher dissimilarity from its full class distribution, whereas class-3 (which is more difficult to learn) benefits from a larger data portion, ensuring adequate coverage of the full class and, thus, preserving the natural performance.

*Table 19.* Similarity of the full dataset and the coresets extracted from poisoned CIFAR10.

| Dataset | $95^{th}$-Percentile Nearest-Neighbor Distance | | | | Maximum Mean Discrepancy | | | |
|---|---|---|---|---|---|---|---|---|
| | BadNets | Blend | IAB | WaNet | BadNets | Blend | IAB | WaNet |
| Full | $0.0112_{\pm 0.0002}$ | $0.0114_{\pm 0.0003}$ | $0.0116_{\pm 0.0002}$ | $0.0113_{\pm 0.0003}$ | $0.0121_{\pm 0.0004}$ | $0.0130_{\pm 0.0006}$ | $0.0142_{\pm 0.0012}$ | $0.0126_{\pm 0.0005}$ |
| Class 0 | $0.0091_{\pm 0.0004}$ | $0.0102_{\pm 0.0026}$ | $0.0104_{\pm 0.0012}$ | $0.0108_{\pm 0.0002}$ | $0.0015_{\pm 0.0008}$ | $0.0027_{\pm 0.0015}$ | $0.0023_{\pm 0.0010}$ | $0.0019_{\pm 0.0002}$ |
| Class 1 | $0.0107_{\pm 0.0004}$ | $0.0111_{\pm 0.0004}$ | $0.0104_{\pm 0.0006}$ | $0.0094_{\pm 0.0006}$ | $0.0297_{\pm 0.0015}$ | $0.0334_{\pm 0.0036}$ | $0.0331_{\pm 0.0062}$ | $0.0096_{\pm 0.0029}$ |
| Class 2 | $0.0114_{\pm 0.0007}$ | $0.0119_{\pm 0.0008}$ | $0.0126_{\pm 0.0010}$ | $0.0120_{\pm 0.0011}$ | $0.0022_{\pm 0.0006}$ | $0.0035_{\pm 0.0009}$ | $0.0032_{\pm 0.0012}$ | $0.0027_{\pm 0.0006}$ |
| Class 3 | $0.0084_{\pm 0.0004}$ | $0.0091_{\pm 0.0003}$ | $0.0076_{\pm 0.0008}$ | $0.0078_{\pm 0.0001}$ | $0.0007_{\pm 0.0003}$ | $0.0008_{\pm 0.0001}$ | $0.0004_{\pm 0.0002}$ | $0.0002_{\pm 0.0001}$ |
| Class 4 | $0.0101_{\pm 0.0011}$ | $0.0103_{\pm 0.0003}$ | $0.0110_{\pm 0.0016}$ | $0.0101_{\pm 0.0010}$ | $0.0039_{\pm 0.0014}$ | $0.0037_{\pm 0.0007}$ | $0.0045_{\pm 0.0010}$ | $0.0023_{\pm 0.0006}$ |
| Class 5 | $0.0111_{\pm 0.0004}$ | $0.0112_{\pm 0.0013}$ | $0.0121_{\pm 0.0010}$ | $0.0109_{\pm 0.0012}$ | $0.0033_{\pm 0.0011}$ | $0.0035_{\pm 0.0009}$ | $0.0037_{\pm 0.0005}$ | $0.0015_{\pm 0.0005}$ |
| Class 6 | $0.0122_{\pm 0.0004}$ | $0.0110_{\pm 0.0008}$ | $0.0120_{\pm 0.0014}$ | $0.0116_{\pm 0.0011}$ | $0.0068_{\pm 0.0003}$ | $0.0039_{\pm 0.0009}$ | $0.0035_{\pm 0.0006}$ | $0.0036_{\pm 0.0007}$ |
| Class 7 | $0.0116_{\pm 0.0008}$ | $0.0113_{\pm 0.0007}$ | $0.0120_{\pm 0.0004}$ | $0.0111_{\pm 0.0007}$ | $0.0161_{\pm 0.0029}$ | $0.0134_{\pm 0.0025}$ | $0.0169_{\pm 0.0032}$ | $0.0135_{\pm 0.0035}$ |
| Class 8 | $0.0142_{\pm 0.0008}$ | $0.0142_{\pm 0.0016}$ | $0.0141_{\pm 0.0016}$ | $0.0149_{\pm 0.0017}$ | $0.0199_{\pm 0.0025}$ | $0.0228_{\pm 0.0050}$ | $0.0214_{\pm 0.0075}$ | $0.0193_{\pm 0.0028}$ |
| Class 9 | $0.0113_{\pm 0.0007}$ | $0.0113_{\pm 0.0004}$ | $0.0108_{\pm 0.0004}$ | $0.0114_{\pm 0.0008}$ | $0.0140_{\pm 0.0012}$ | $0.0116_{\pm 0.0015}$ | $0.0111_{\pm 0.0020}$ | $0.0078_{\pm 0.0026}$ |

