# OpenReview forum: "Anti-Backdoor Coreset Selection via Cumulative Entropy"
_ICML.cc/2026/Conference — ICML 2026 regular_

### Official Review · Reviewer_JN2b · 2026-03-09

**Soundness:** 3
**Presentation:** 3
**Significance:** 3
**Originality:** 2
**Overall Recommendation:** 4
**Confidence:** 4

**Summary:**

The paper proposes a training-time backdoor mitigation method, ABCS. ABCS introduces Cumulative Entropy to detect samples at different training stages, extracting information-rich and low-harm coresets. ABCS demonstrates performance exceeding state-of-the-art (SOTA) defense methods on various datasets and models.

**Compliance With Llm Reviewing Policy:**

Affirmed.

**Final Justification:**

Thank the authors for their response, which has fully addressed my concerns. The paper is clearly written and supported by a comprehensive experimental evaluation. Considering that the scope of the work remains within the relatively narrow area of backdoor mitigation for classification models, I choose to maintain my final score at Weak Accept.

**Key Questions For Authors:**

1. For small-scale models, even if the coreset size is reduced relative to the original dataset, theoretically it shouldn't significantly impact the performance after training. If the model scale increases (such as ResNet-101 or ViT), will a smaller coreset affect the average accuracy under benign performance?

2. The essence of ABCS is to obtain a clean and information-rich coreset. Can this coreset be applied to tasks beyond classification (such as image captioning, segmentation, etc.)?

3. If the adversary understands the defense mechanism of ABCS, and trains a trigger pattern under similar coreset extraction experimental settings, will this affect the defense effectiveness of ABCS?

**Limitations:**

yes

**Strengths And Weaknesses:**

Strengths:
1. The paper has a clear structure and sufficient experiments. Each step of the method is supported by theoretical analysis and experimental proof, demonstrating strong theoretical rigor and soundness.
2. ABCS achieves performance exceeding state-of-the-art methods and effectively resists various existing attacks.
3. The experimental settings are clearly explained, and the parameter analysis and ablation studies are comprehensive, demonstrating strong reproducibility.

Weaknesses:

[Major]
1. The models used in the experiments are generally small-scale, and larger models such as transformer-based models are not employed. The possibility of overfitting when larger models are trained on small coresets is not fully discussed.
2. Compared to utilizing both malicious and clean samples simultaneously, ABCS only collected a coreset composed of clean samples, theoretically suppressing the application value of a large number of unselected samples.
3. The impact of adaptive attacks on ABCS was not discussed.

[Minor]
1. The subject of the second sentence (the third row) in the Threat Model Section is incorrectly 'She'.

---

> ### Author Rebuttal · Authors · 2026-03-31
>
> We thank the reviewer for the feedback and appreciate the positive sentiment.
> Below, we provide responses to all mentioned weaknesses individually.
> Please, note that results of ACC, ASR, $\rho$ and $r_{se}$ are shown in percent.
>
> ### Response to W1 & Q1:
>
> Using a model with larger capacity, such as ResNet101, slightly increases the natural accuracy.
> Below, we summarize the experiments with ResNet101, where the naive training on the clean dataset yields ACC of 95.37% and 63.14% for CIFAR-10 and Tiny-ImageNet, respectively.
> Overall, the larger model capacity does not affect ABCS's defense.
>
> We observe that the selected coreset is adaptively enlarged for CIFAR10, while it remains similar for Tiny-ImageNet compared to our main results in Table 4 with ResNet18 and ResNet34.
> Most importantly, the backdoor elimination remains highly effective and also the yielded ACC is comparable to clean-dataset training.
> Overfitting is not an issue in this setting per-se.
>
> |Datataset   | Attack | ACC   | ASR  | $r_{se}$ | $\rho_{bng}$ |
> |------------|--------|-------|------|----------|--------------|
> | CIFAR10    | BadNet | 95.03 | 0.74 | 56.83    | 0.00 |
> |            | Blend  | 95.11 | 0.54 | 57.28    | 0.00 |
> |            | IAB    | 95.31 | 1.84 | 62.29    | 0.01 |
> |            | WaNet  | 95.25 | 0.83 | 62.16    | 0.29 |
> |TinyImageNet| BadNet | 62.96 | 0.05 | 81.29    | 0.00 |
> |            | Blend  | 62.75 | 0.59 | 81.41    | 0.03 |
> |            | IAB    | 62.26 | 0.14 | 84.74    | 0.01 |
> |            | WaNet  | 62.03 | 0.35 | 82.11    | 0.06 |
>
> ### Response to W2:
>
> While coreset selection removes data sample it does not sacrifice information.
> An informative coreset comprises high-impact samples that collectively represent the full dataset, **covering the information of unselected samples**.
> As shown in our coreset coverage analysis (Section F), the selected coreset covers the latent space distribution of the full dataset.
> Thus, training on it preserves the dataset's information while reducing computational cost.
>
> From a defense perspective, prior methods such as V&B, Harvey, and DBD train on as many clean samples as possible, requiring highly accurate benign/poisoned splitting.
> Mis-splitting may reintroduce the backdoor and is difficult to recover from, leading to failures when encountering hard attacks such as LowFrequency and Adaptive Blend.
> ABCS instead extracts an informative coreset that covers the full dataset information, largely eliminating the need for highly precise separation of all benign and poisoned samples in dataset splitting.
>
> ### Response to Q2:
>
> While our current evaluation focuses on classification tasks, the idea of cumulating model's prediction uncertainty is generalizable to other supervised learning settings.
> Exploring its application to tasks such as segmentation or image captioning is a promising direction that we leave for future work.
>
> ### Response to W3 & Q3:
>
> We agree that studying adaptive attacks is crucial and have thus conduct evaluation on four adaptive attacks in **Section E** in the appendix of our paper:
> 1. one-shot poisoning guided by sorting of CENT value for all benign samples (Section E.1),
> 2. single-class poisoning to reduce backdoor significance (Section E.2),
> 3. randomized trigger assignment across four patterns (Section E.3), and
> 4. label randomization of poisoned samples to lower the effective poisoning ratio and increase learning difficulty (Section E.4).
>
> ABCS is highly robust against the first three attacks.
> However, it is affected by "label randomization" (Section E.4), but the attack is also less effective:
> Aggressive label randomization (50%) sacrifices the backdoor's ASR for a higher learning difficulty (cf. Figure 18) in an attempt to bypass ABCS.
> Indeed, while ABCS lowers the ASR (cf. Table 14), it does not suppress it to the very minimum.
> For less aggressive label randomization (30% -> 10%) and thus more effective attacks, ABCS also improves.
>
> ### Response to Minor Weakness:
>
> Thank you very much for bringing this up.
> We are happy to opt for a general-neutral phrasing instead:
> > We consider backdoors that are introduced via data poisoning, that is, **the adversaries** manipulate samples from a training dataset D. However, **they** cannot influence the training process.

---

> > ### Author Rebuttal · Reviewer_JN2b · 2026-04-02
> >
> > The authors have addressed all of my concerns. Considering the overall quality and completeness of the paper, I will maintain my original score. At the same time, I suggest moving some of the more important adaptive attack results from the appendix to the main paper.

---

> > > ### Author Response · Authors · 2026-04-04
> > >
> > > We appreciate the reviewer's acknowledgement of our rebuttal.
> > > In our revision, we will include the most important adaptive attacks, that we currently discuss in the appendix, in the main part of our paper already.

---

### Official Review · Reviewer_pxFC · 2026-03-09

**Soundness:** 3
**Presentation:** 3
**Significance:** 4
**Originality:** 3
**Overall Recommendation:** 5
**Confidence:** 5

**Summary:**

This paper proposes ABCS, a training-time defense against the backdoor attack. Instead of explicitly splitting poisoned and benign samples, the method reframes defense as a coreset selection problem: it aims to identify a smaller subset of informative and benign samples from a poisoned training set using cumulative entropy, then retrains a model from scratch on this subset. The paper shows strong results across several standard image benchmarks and attacks, with low ASR and good clean accuracy.

**Compliance With Llm Reviewing Policy:**

Affirmed.

**Final Justification:**

The paper considering training-time backdoor defense as a coreset selection problem is a promising direction. The paper is well-organised and easy to follow. I think my concerns are fully addressed. I keep positive about this paper.

**Key Questions For Authors:**

- I notice that ABCS involves many hyperparameters, and the paper provides extensive analysis of them, including the warm-up epochs, selection epochs, label-smoothing strength, and regularization strength. One question I still have is whether the poisoning ratio of the attacker affects the appropriate hyperparameter settings for ABCS. In particular, if the poisoning ratio changes, would the optimal choices of these parameters also need to change?

- ABCS is built on the assumption that **poisoned samples** are easier to learn and therefore have lower cumulative entropy, while **benign informative samples** remain more uncertain and get higher CENT. My question is: If the ABCS is used in some easy task like MINIST, where the clean samples are already very easy and distinctive, while some injected backdoor patterns are hard to learn, leading to low entropy on clean samples. If that happens, can the ABCS still separate clean and poisoned samples well by CENT?

**Limitations:**

Yes. The authors discussed the limitations and potential negative social impact of their work.

**Strengths And Weaknesses:**

**Strengths**：
- Framing training-time backdoor defense as a coreset selection problem is a promising direction. Compared with traditional training-time defenses that aim to explicitly split the entire dataset into clean and poisoned subsets, ABCS avoids requiring precise dataset separation, which makes the approach more attractive in realistic settings.

- The definition of cumulative entropy is intuitive and reasonably motivated. The core idea is that poisoned samples often share similar trigger patterns and are learned more easily, causing them to exhibit lower cumulative entropy than informative benign samples.

- The paper includes a fairly comprehensive empirical evaluation, covering multiple attacks, including both standard static backdoor attacks and adaptive variants, along with ablation studies and comparisons against several baselines. The results show that ABCS achieves strong robustness with lower ASR while maintaining competitive clean accuracy.

**Weaknesses**:
- The overall pipeline is not presented clearly enough. In particular, the role of the warm-up stage is confusing. From the method description, ABCS **resumes training** the model for $T_{se}$ epochs in order to learn the entropy distribution of all samples and separate uncertain samples from poisoned ones. Under this description, it is unclear what additional role the warm-up stage plays in the final coreset selection process. Moreover, in Step 3, the cumulative entropy of each sample appears to be computed based on the training procedure in Step 2 rather than Step 1. This makes the relation between the warm-up stage and the actual coreset-selection stage hard to follow. The paper should clarify whether the model is first trained for $T_{wa}$ warm-up epochs and then further trained for $T_{se}$ epochs to select the coreset, or whether Step 1 is mainly used to estimate the threshold $\tau$.

- Another major concern is that ABCS intentionally trains the final clean model only on a reduced subset $D_{bng}$, which is much smaller than the original datasets. In particular, I notice that the selection rate for some classes can be as low as 0.05. On more complex tasks or highly imbalanced datasets, such a low selection rate may lead to insufficient class coverage and consequently hurt clean performance, even introduce bias or cause an OOD problem, which would weaken the utility of the deployed model. This concern is partly reflected in the paper: in Figure 11, the accuracy of some classes (e.g., class 27) drops to around 50%. Therefore, the paper should more explicitly discuss whether ABCS is fundamentally limited in imbalanced settings. If so, it would also be important to clarify under what data conditions or application scenarios ABCS is expected to remain reliable.

- (Minor concerns:) Notation/formulation around the selection ratio is inconsistent: In section 2, the selection ratio is defined as $|D_{bng}| / |D_c|$ while in section 4.1, the selection ratio is defined as $|D_{bng}|/|D|$;

---

> ### Author Rebuttal · Authors · 2026-03-31
>
> We appreciate the reviewer's positive feedback and the constructive comments. Below, we address each concern individually. Results of ACC, ASR, $\rho$ and $r_{se}$ are shown in [%].
>
> ### Response to W1:
>
> Our apologies that the paper has not been clear on this implementation, which we will clarify in our revision.
>
> * In the warm-up phase (Section 4.1), we train for $T_{wa}$ epochs to establish knowledge of both benign and poisoned samples and compute CENT to determine the coreset size $s$.
>
> * We then refresh CENT computing and resume fine-tuning the "warmed-up model" for $T_{se}$ epochs as detailed in Section 4.2.
> Each epoch goes through three steps, whereas in *Step-3*, we compute the samples' normalized entropy AND cumulate it with results from all previous epochs yielding CENT after this particular epoch.
> This CENT values allows us to gather $s$ informative samples after each epoch.
>
> * However,---and we believe this is the source of the misunderstanding---only the final selection after $T_{se}$ epochs is then used for training the final model as described in Section 4.3.
>
> Warm-up training accounts for attacks' diverse learning difficulties.
> For example, BadNet is learned rapidly, while WaNet is more slowly.
> As shown below, the positive gap of normalized entropy $\Delta H = H_{bng}-H_{poi}$, beneficial for data separation, emerges for WaNet only after several training epochs.
>
> |Metric|Attack|Epoch-1|Epoch-3|Epoch-5|Epoch-7|Epoch-9|
> |-|-|-|-|-|-|-|
> |$\Delta H$|BadNet|+0.46|+0.27|+0.24|+0.19|+0.15|+0.11|
> ||WaNet|-0.02|-0.06|-0.13|-0.01|+0.05|+0.08|
> |ASR|BadNet|98.43|100.00|100.00|100.00|100.00|
> ||WaNet|6.25|0.78|12.50|57.81|86.72|
>
> ### Response to W2:
>
> This is a great observation.
> Please note, though, that the class-wise selection ratio does not reflect "class coverage."
> As an extreme example, a class represented by 1000 times the exact same sample, can be reduced to 1 sample w/o loosing informativeness.
>
> ABCS's coreset selection is driven by sample informativeness rather than class-wise data quantity.
> Classes that are inherently hard-to-learn or rare in the dataset have high prediction uncertainty during training, increasing their likelihood of being selected rather than exclusion.
>
> * Using CIFAR-10 as already balanced example, training on the extracted coreset yields comparable accuracy to training on the full dataset (cf. Figure 6(a)).
> * In contrast, GTSRB is highly imbalanced: classes 21 and 27 have small proportions (0.008 and 0.006) and low baseline accuracy (approx. 70% and 50%).
> After coreset selection under Blend attack (blue line in Figure 11), their proportions only slightly shift to 0.004 and up to 0.009, and the per-class accuracy remains comparable to training on the full dataset (black line in Figure 11).
> This results shows that coreset selection adjusts to data coverage, particularly for minority and hard classes.
> Also, the drop for class 27 is inherently more severe for natively training the full clean dataset than for ABCS against Blend attack.
>
> ABCS thus is not fundamentally limited in imbalanced settings, but adapts dynamically.
> We will clarify in the final version of the paper.
>
> ### Response to W3:
>
> Thank you. We will revise by consistently using $|D_{bng}|/|D|$.
>
> ### Response to Q1:
>
> Overall, ABCS targets benign coreset selection that reliably excludes poisoned samples regardless of the attack strategy.
> Thus, default hyperparameters remain broadly effective.
>
> Below, we test with the poisoning rate of 0.01 under Blend and WaNet attacks.
> Results are reported in ACC/ASR.
> Longer warm-up and selection periods ($T_{wa}$, $T_{se}$) make ABCS more robust.
> Increasing the unlearning parameters $\gamma$ and $\epsilon$ also improves ABCS's defence, while overly large values may have a marginal impact.
>
> - $T_{se}$:
>
> |Attack|10|20|30|40|50|
> |-|-|-|-|-|-|
> |Blend|94.78/17.40|94.34/11.64|94.83/5.67|94.68/0.90|94.55/1.31|
> |WaNet|94.50/50.60|94.17/8.52 |94.31/7.10|94.24/4.15|94.08/4.86|
>
> - $T_{wa}$:
>
> |Attack|3|5|8|10|20|
> |-|-|-|-|-|-|
> |Blend|94.72/25.16|94.75/22.74|94.46/3.12|94.68/0.90|94.23/1.22|
> |WaNet|94.15/44.58|94.89/37.46|94.62/5.36|94.24/4.15|94.19/6.84|
>
> - Label smoothing factor $\epsilon$:
>
> |Attack|0.3|0.5|0.7|0.9|0.99|
> |-|-|-|-|-|-|
> |Blend|94.22/23.53|94.84/18.34|94.90/2.11|94.68/0.90|94.72/1.68|
> |WaNet|94.27/50.71|94.06/30.22|94.75/5.70|94.24/4.15|94.41/4.29|
>
> - Unlearning factor $\gamma$:
>
> |Attack|0.01|0.05|0.1|0.5|1.0|
> |-|-|-|-|-|-|
> |Blend|94.35/18.82|94.77/8.66|94.68/0.90|94.43/1.21|94.75/1.31|
> |WaNet|94.89/58.09|94.10/7.21|94.24/4.15|93.99/7.08|94.47/5.01|
>
> ### Response to Q2:
>
> Training on clean MNIST yields 99.22% ACC with VGG11.
> Below, we test on BadNet and Blend attacks with using grayscale triggers.
> ABCS adaptively reduces the coreset size and effectively eliminates poisoned samples, showing its robustness also for the easy task of MNIST.
>
> |Attack|ACC|ASR|$\rho_{bng}$|$r_{se}$|
> |-|-|-|-|-|
> |BadNet|99.16|0.00|0.00|25.50|
> |Blend|99.14|0.16|0.07|24.64|

---

> > ### Author Rebuttal · Reviewer_pxFC · 2026-04-02
> >
> > Thank the authors for addressing my concerns. I have increased my score and remain positive about this paper.

---

> > > ### Author Response · Authors · 2026-04-04
> > >
> > > We sincerely thank the reviewer for the positive feedback and the adjusted score.
> > > We will revise the paper to address the reviewer's comments and ensure clarity of ABCS's implementation.

---

### Official Review · Reviewer_gUty · 2026-03-09

**Soundness:** 3
**Presentation:** 3
**Significance:** 2
**Originality:** 2
**Overall Recommendation:** 4
**Confidence:** 4

**Summary:**

This paper proposes using training dynamics or uncertainty to filter out poisoned samples crafted by backdoor attacks. The authors proposed a multi-stage training approach to amplify the uncertainty gap between benign and poisoned samples, enabling easy filtering of poisoned samples. Experimental studies show that the proposed method outperforms the multiple defense baselines across different attacks and settings.

**Compliance With Llm Reviewing Policy:**

Affirmed.

**Final Justification:**

The authors have addressed my major concerns.

**Key Questions For Authors:**

N/A

**Limitations:**

yes

**Strengths And Weaknesses:**

**Strengths**
1. The presentation of the method is easy to follow.
2. The proposed method achieves the best results across different settings compared to existing baselines.
3. Multiple in-depth analyses were conducted to support the effectiveness of the proposed method from different angles.

**Weaknesses**
1. Training dynamics (or a similar idea) has been studied in the backdoor defense literature (see [2]), Although the design is slightly different, the underlying idea is largely the same. Thus, I suggested that the authors at least compare their work with the related approach. In addition, prior work [1] has shown that selecting 50% of the uncertain samples from the full training set can achieve performance comparable to training on the entire dataset. Thus, this observation itself is not novel to the community.

2. Although the paper focuses on the training-time defense, it would be nice to compare it to some test-stage defenses to verify the effectiveness of their approach.

3. The equation on line 273 is quite confusing. I assume that the goal here is to minimize the loss, but line 269 uses "unlearn", which may lead readers to interpret the objective as removing the model’s knowledge of the target set. Instead, the goal here is to penalize confidence via label smoothing, leading to increased uncertainty (correct me if I misunderstand; then, the narrative should be updated to avoid confusion).

4. Since the defense removes only suspected poisoned samples from the training set, if the backdoor is fully removed, the model should behave similarly to a benign classifier on these inputs. However, the reported ASR in Table 10 (1.03% in SST-2 with 92% accuracy and 0.31% in the 5-class setting with 48% accuracy) seems unexpectedly low, especially in the multi-class case, where misclassifications would typically spread across non-target classes. In short, even though the backdoor has been fully removed, it’s unlikely (based on my previous experience ) that ASR can reach 0.31% due to the systematic error (i.e., 48% accuracy). This raises concerns about potential evaluation artifacts or target-class suppression. I suggest clarifying the exact ASR definition, reporting the number of triggered test samples, and providing confusion matrices after defense. Meanwhile, I would like to see the ASR of the benign models for both CV and NLP tasks, which can serve as an approximate lower bound.

**References**
1. Swayamdipta et al 2020. Dataset Cartography: Mapping and Diagnosing Datasets With Training Dynamics
2. He et al. 2024. SEEP: Training Dynamics Grounded Latent Representation Search for Mitigating Backdoor Poisoning Attacks

---

> ### Author Rebuttal · Authors · 2026-03-31
>
> We sincerely thank the reviewer for the feedback. Below, we respond to raised weaknesses. Results of ACC, ASR, $\rho$ and $r_{se}$ are shown in [%].
>
> ### Response to W1:
>
> Indeed, (1) training dynamics have been studied for backdoor defense, and (2) also 50% of random selection has been demonstrated before. However,
>
> **1. Training dynamics.**
> The core design of SEEP and ABCS differ fundamentally in this regard, although both share similar defense objectives.
> ABCS explicitly accumulates training dynamics to extract a benign and representative coreset, while SEEP primarily identifies poisoned samples with high precision, not accounting for the informativeness of remaining samples.
>
> We would have loved to compare to SEEP in rebuttal. Unfortunately, the authors does not provided an executable code implementation :(
> Thus, we compare to results reported in the paper by following the same attack setup.
> ABCS remains comparablely robust as SEEP against attacks on NLP tasks.
>
> |Attack|Defense|ACC|ASR|
> |-|-|-|-|
> |BadNet|SEEP|**92.6**|7.4|
> ||ABCS|92.3|**4.9**|
> |Synactic|SEEP|91.5|**10.0**|
> ||ABCS|**91.7**|12.7|
>
> **2. Selection rate of 50%.**
> Please note that prior work focuses on natural accuracy only, not considering backdoors at all.
> **Figures 2 and 9** demonstrate this relation to underscore the importance of an adjusted selection.
> We do not present a novel contribution in minimizing coreset size, but a criterion that enables excluding poisonous samples.
>
> Moreover, ABCS is not positioned as a universally optimal defense; instead, we propose a new defense paradigm.
>
> ### Response to W2:
>
> Thanks for the suggestion, that we gladly put into practice:
> We evaluate two post-train defenses that aim to erase the backdoor from a trained model, ANP [1] and SAU [2].
> Due to the short time, we consider CIFAR10 only, but are happy to extend to other datasets for the final version.
>
> |Attack|ANP|SAU|ABCS|
> |---|---|---|---|
> |BadNet|92.08/1.27|92.97/2.72|**94.38**/**1.00**|
> |Blend|88.55/4.40|91.64/1.20|**94.62**/**0.77**|
> |IAB|93.08/2.08|**94.91**/**0.45**|94.30/1.22|
> |WaNet|92.48/**0.60**|93.11/1.81|**94.13**/0.89|
>
> [1] Wei et al., "Shared adversarial unlearning: Backdoor mitigation by unlearning shared adversarial examples," NeurIPS 2023.
>
> [2] Wu et al., "Adversarial Neuron Pruning Purifies Backdoored Deep Models," NeurIPS 2021.
>
> ### Response to W3:
>
> Our apologies for the confusion.
> Rather than removing knowledge of samples in $D_{ul}$, we indeed minimize
> $L_{ul}$
> with their smoothed/softend labels, which lowers and increases the expected confidence for the ground-truth and non-ground-truth classes for a more uniform distribution. This enlarges prediction uncertainty for $D_{ul}$, thereby "unlearning" them.
>
> ### Response to W4:
>
> We compute ASR only with using test samples from non-target classes; By default, the target class is 0.
> In this setting, we obtain ACC/ASR of 92.32/5.41 and 50.77/3.53 after completing the fine-tuning on clean SST-2 and SST-5, respectively.
> Below, we use SST-5 with BERT-base to help shed light on the reviewer's concerns.
>
> ABCS indeed produces a slight class distribution shift.
> As BERT is pre-trained, classes with more samples dominate the initial fine-tuning.
> The correlation matrices show a large imbalance with major decrease for class 0 at the highest accuracy (*ACC=48.41%*):
>
> |Label\Predicted|Class-0|Class-1|Class-2|Class-3|Class-4|
> |-|-|-|-|-|-|
> |Class-0|10.07|86.33|00.00|03.60|00.00|
> |Class-1|03.81|80.28|02.42|13.49|00.00|
> |Class-2|01.31|50.22|06.99|39.30|02.18|
> |Class-3|00.36|11.11|01.79|69.18|17.56|
> |Class-4|00.00|03.64|01.21|47.88|47.27|
>
> Such imbalance also suppresses predictions to the target class 0:
>
> |Label\Predicted|Class-0|Class-1|Class-2|Class-3|Class-4|
> |-|-|-|-|-|-|
> |Class-0|00.62|41.06|05.72|42.10|10.50|
>
> After fine-tuning, in turn, the matric is better balanced, but with a slight ACC reduction to *47.87%*:
>
> |Label\Predicted|Class-0|Class-1|Class-2|Class-3|Class-4|
> |-|-|-|-|-|-|
> |Class-0|37.41|43.88|14.39|02.88|01.44|
> |Class-1|14.19|53.63|21.80|10.03|00.35|
> |Class-2|02.62|20.96|28.38|40.61|07.42|
> |Class-3|00.36|03.94|07.89|50.18|37.63|
> |Class-4|00.00|01.21|01.21|27.88|69.70|
>
> Attack success slightly recovers, though:
>
> |Label\Predicted|Class-0|Class-1|Class-2|Class-3|Class-4|
> |-|-|-|-|-|-|
> |Class-0|03.43|21.31|19.33|38.15|17.78|
>
> **We suggest to provide results after the complete fine-tuning in our revision.**
>
> For SST-2, this distribution shift appears as well, leading to class-wise imbalance at the highest ACC.
> After completing fine-tuning, we obtain ACC/ASR = 91.74/5.86.
> We further test poisoning with target class 1, yielding ACC/ASR = 91.63/6.54.
>
> Finally, we show the ASR of benign models for CV tasks:
>
> |Dataset|Attack|ASR|
> |-|-|-|
> |CIFAR10|BadNets|0.70%|
> ||Blend|0.79%|
> ||CLB|0.86%|
> ||IAB|1.76%|
> ||WaNet|0.53%|
> ||ISSBA|1.02%|
> ||LF|2.51%|
> ||A-Blend|3.44%|
> |Tiny-ImageNet|BadNets|0.02%|
> ||Blend|0.10%|
> ||IAB|0.16%|
> ||WaNet|0.31%|
> ||ISSBA|0.47%|
> ||LF|0.20%|
> ||A-Blend|0.33%|

---

> > ### Author Rebuttal · Reviewer_gUty · 2026-04-04
> >
> > Thanks for the detailed response. Given that the ASR of benign models on SST2/SST5 is much higher than the ones from your defense. I would like to hear more explanation about this phenomenon.

---

> > > ### Author Response · Authors · 2026-04-04
> > >
> > > Thank you for the response and our apologies for the brevity of our initial interpretation, which was influenced by the limited space available in the rebuttal and our paper's focus on the vision domain.
> > > However, we provide a more detailed explanation of the NLP task below.
> > >
> > > First, we compare the class ratios in the full dataset and the coreset selected by ABCS.
> > > Note that each ratio is expressed in percent in the following tables.
> > >
> > > * **Ratios per class for SST-2:**
> > >
> > > |Dataset|Class-0 (Positive)|Class-1 (Negative)|
> > > |-|-|-|
> > > |Full|44.22|55.78|
> > > |Coreset|30.76|69.24|
> > >
> > > * **Ratios per class for SST-5:**
> > >
> > > |Dataset|Class-0|Class-1|Class-2|Class-3|Class-4|
> > > |-|-|-|-|-|-|
> > > |Full|12.78|25.96|19.01|27.18|15.07|
> > > |Coreset|6.09|34.85|16.04|28.07|14.94|
> > >
> > > The selected coresets exhibit a noticeable distribution shift compared to the original distribution of SST-2 and SST-5.
> > >
> > > ### SST-5
> > >
> > > For SST-5, the proportion of the target class (class 0) is significantly reduced, leading to very low accuracy at the early stages of fine-tuning.
> > > This shift biases the model toward high-quantity classes (i.e. classes 1 and 3), resulting in substantially higher accuracy for these classes than for others.
> > >
> > > This behavior does not indicate effective backdoor suppression yet, though.
> > > Instead, it reflects insufficient learning of class 0, causing even poisoned samples to be misclassified.
> > > By averaging across all classes, the strong performance on classes 1 and 3 lead to an overall high ACC, masking the class imbalance and highlighting **the necessity of completing fine-tuning**.
> > > After full fine-tuning, the model achieves a more balanced class-wise performance (although there is a slight drop in overall ACC).
> > >
> > > > **We thus suggest to provide results after the complete fine-tuning in our revision, rather than in the beginning as done in the original version of the paper.**
> > >
> > > Please refer to the analysis and specific values for SST-5 in the initial response.
> > > Next, we proceed to interpret the issue in SST-2.
> > >
> > > ### SST-2
> > >
> > > A similar shift is observed for SST-2, where early fine-tuning is biased toward class 1.
> > > Consequently, *Negative* samples are less likely to be misclassified as *Positive*.
> > > Since the final coreset is nearly free of poisoned samples (cf. Table 10, $\rho_{bng} \approx 0.0%$), poisoned samples with inherently negative sentiment are correctly classified as *Negative*.
> > > Below, we present the ACC/ASR matrices with using ABCS when (A) selecting the best accuracy *(as done in the initial paper submission)* or (B) completing fine-tuning on the coreset *(as now suggested to provide a more accurate representation)* to explain the unexpectedly low ASR.
> > >
> > > **A. Selecting the best accuracy:**
> > >
> > > - Clean test (ACC = 92.20%)
> > >
> > > |Label\Predicted|Positive|Negative|
> > > |-|-|-|
> > > |Positive|86.03|14.97|
> > > |Negative|1.40|98.60|
> > >
> > > - Poisoned test (ASR = 1.64%)
> > >
> > > |Label\Predicted|Positive|Negative|
> > > |-|-|-|
> > > |Positive|1.64|98.36|
> > >
> > > **B. Completing fine-tuning on the coreset:**
> > > After completing fine-tuning, in turn, the class-wise accuracy becomes more balanced, while the averaged ACC for SST-2 slightly drops.
> > >
> > > - Clean test (ACC = 91.74%)
> > >
> > > |Label\Predicted|Positive|Negative|
> > > |-|-|-|
> > > |Positive|89.41|10.59|
> > > |Negative|5.84|94.16|
> > >
> > > - Poisoned test (ASR = 5.86%)
> > >
> > > |Label\Predicted|Positive|Negative|
> > > |-|-|-|
> > > |Positive|5.86|94.14|
> > >
> > >
> > > ### Summary
> > >
> > > ABCS results in a shift of the class-wise distribution compared to the original dataset.
> > > Since the BERT model is pre-trained, this distribution shift makes the model strongly converge to classes with larger number of samples, which is particularly true in the beginning of fine-tuning.
> > > After the convergence to the dominant class(es), the model starts learning other classes as well, so that it eventually yields a more balanced accuracy distribution across all classes.
> > >
> > > Importantly, this does not indicate that the coreset introduces a high class-wise imbalance in order to preserve the accuracy.
> > > Instead, the coresets isolated by ABCS maintain a similar accuracy distribution compared to the clean fine-tuning results (cf. below)
> > >
> > > **ACC/ASR after fine-tuning on clean datasets:**
> > >
> > > - **SST-2:** Clean test (ACC=92.32)
> > >
> > > |Label\Predicted|Positive|Negative|
> > > |-|-|-|
> > > |Positive|89.25|10.75|
> > > |Negative|4.73|95.27|
> > >
> > > - **SST-2:** Poisoned test (ASR=5.41)
> > >
> > > |Label\Predicted|Positive|Negative|
> > > |-|-|-|
> > > |Positive|5.41|94.59|
> > >
> > > - **SST-5:** Clean test (ACC=50.77)
> > >
> > > |Label\Predicted|Class-0|Class-1|Class-2|Class-3|Class-4|
> > > |-|-|-|-|-|-|
> > > |Class-0|47.48|45.32|5.04|2.16|0.00|
> > > |Class-1|17.65|59.52|14.88|6.92|1.03|
> > > |Class-2|3.49|33.19|23.14|35.81|4.37|
> > > |Class-3|0.36|3.23|10.39|58.06|27.96|
> > > |Class-4|0.61|0.00|1.82|33.33|64.24|
> > >
> > > - **SST-5:** Poisoned test (ASR=3.53)
> > >
> > > |Label\Predicted|Class-0|Class-1|Class-2|Class-3|Class-4|
> > > |-|-|-|-|-|-|
> > > |Class-0|3.53|21.83|28.90|37.01|8.73|

---

### Official Review · Reviewer_faaf · 2026-03-10

**Soundness:** 2
**Presentation:** 3
**Significance:** 2
**Originality:** 2
**Overall Recommendation:** 4
**Confidence:** 3

**Summary:**

This paper transforms backdoor defense into a coreset selection problem. Based on the characteristics of low prediction uncertainty and low frequency associated with poisoned samples, this paper proposes the ABCS. It utilizes cumulative entropy as a criterion to evaluate the dynamic informativeness of samples, and incorporates unlearning and label smoothing techniques to further amplify the disparity between benign and poisoned samples. Experiments demonstrate that ABCS can accurately extract a benign coreset from poisoned data and that training models on this coreset not only withstands complex backdoor attacks but also maintains natural performance. Moreover, this method requires no prior clean data and incurs low computational overhead, making it highly practical.

**Compliance With Llm Reviewing Policy:**

Affirmed.

**Final Justification:**

The auhtors addressed part of my concerns, so I have increased the score to 4.

**Key Questions For Authors:**

1. Is it possible to establish a theoretical framework to ensure the defense does not completely collapse?

2. Do you have any specific strategies to mitigate the decline in natural accuracy?

**Limitations:**

Yes

**Strengths And Weaknesses:**

Strengths:

This paper makes certain contributions. Firstly, without relying on any clean reference dataset, the method can effectively defend against various backdoor attacks of different levels of difficulty, while maintaining the natural accuracy. Secondly, the method extracts a benign coreset for the final training from scratch, and the overall training time does not increase, significantly improving computational efficiency. Finally, the method remains effective even when the dataset is clean, demonstrating strong adaptability and avoiding the issue of model performance degradation.

Weaknesses:

1. The defense hinges on the assumption that backdoor samples exhibit lower prediction uncertainty than benign samples. The proposed mechanism to unlearn uncertain samples lacks formal theoretical justification, leaving its reliability unverified when the initial assumption fails.

2. While four adaptive attacks were tested, the evaluation lacks a high-potency, optimization-based attack. Including such vectors is essential to provide a more rigorous and reliable assessment of the defense's robustness.

3. The paper is predominantly empirical. Without a theoretical framework to establish fundamental guarantees, the defense remains vulnerable to worst-case scenarios that could lead to a total breakdown in practice.

4. The method struggles with complex, low-redundancy tasks, showing significant declines in selection ratio and natural accuracy. This limits its viability for large-scale, real-world deployments compared to simple dataset benchmarks.

---

> ### Author Rebuttal · Authors · 2026-03-31
>
> We thank the reviewer for the feedback and respond to all weaknesses and questions raised below.
>
> ### Response to Q1 (W1+W3):
>
> While we do not include a theoretical framework, we build upon existing theoretical work.
> We unlearn through **label-smoothed training**, which comprises two loss terms [2]:
> 1. ground-truth label: $(1-\epsilon+\frac{\epsilon}{C})\cdot\mathcal{L}_{ce}(x,y,\theta)$,
> 2. non-ground-truth labels: $\frac{\epsilon}{C}\sum_{y' \neq y}\mathcal{L}_{ce}(x,y',\theta)$.
>
> Doing so lowers the expected confidence of the ground-truth and distributes the remaining confidence uniformly across other classes.
> Since samples to unlearn, $D_{ul}$, have first been learnt with one-hot labels, the loss for ground-truth labels makes the model forget these samples [1], and the loss for non-ground-truth labels further suppresses the ground-truth confidence [2].
> Label smoothing thus increases a sample's prediction uncertainty rather than aggressively unlearning them with gradient ascent.
>
> We evaluate ABCS across various datasets and attacks.
> Note that the low-uncertainty assumption of poisoned samples does **not** always hold, e.g., for WaNet.
> Without unlearning, the coreset would retain poisoning samples with **high** uncertainty.
> Thus, WaNet is indeed notoriously difficult to defend.
> Since the majority of samples in $D_{ul}$ are benign, label-smoothed training (unlearning) degrades ACC more than ASR, enlarging benign samples' uncertainty and thus reducing $\rho_{bng}$.
>
> ||Metric|5|10|15|20|25|30|35|40|
> |-|-|-|-|-|-|-|-|-|-|
> |w/o Unlearn|$\rho_{bng}$|4.3|4.1|1.5|2.7|1.5|3.2|2.0|1.1|
> |w/ Unlearn|$\rho_{bng}$|1.1|0.7|0.4|0.2|0.2|0.1|0.1|0.1|
> ||$\Delta$ ACC|−7.0|−12.7|−3.9|−2.4|−15.0|−18.0|−10.3|−8.9|
> ||$\Delta$ ASR|+7.0|+6.3|+5.5|−0.8|−0.3|+0.9|−0.8|+0.6|
>
> Moreover, among the four adaptive attacks we present in Section E, **label randomization** can be seen as a potential worst-case adversary that directly targets ABCS's core assumption.
> Noisy labels are known to increase learning difficulty [3] and predictive uncertainty [4,5].
> Figure 18 shows that a high randomization ratio substantially raises the entropy of poisoned samples relative to benign ones, making ABCS's performance degrade (cf. Table 14), but it also dramatically reduces the ASR of backdoor.
> When the adversary decreases the randomization for a stronger attack, also ABCS improves and can substantially reduce ASR.
>
> ### Response to W2:
>
> Thank you for this suggestion.
> In addition, we consider an attack that performs a greedy search to craft poisoned samples with higher uncertainty than random sampling:
>
> *Input: clean dataset $D$, a ResNet18 as $f_{\theta}$, the number of iteration $T$ and target poisoning rate* $\rho$
>
> *Output: a poisoned dataset $\hat{D}$*
>
> *Initialisation:*
> - *Train* $f_{\theta}$ *for 40 epochs on $D$ and poison 50% samples with highest CENT as* $\hat{D}^{(0)}$
> - *Compute step size* $n=\text{clip}(\frac{(0.5-\rho)\cdot|D|}{T})$
>
> *For* $t$ *in* [1,...,$T$]
>
> *Do*
>
> ** *Step 1: Train $f_{\theta}$ from scratch for 40 epochs*
>
> ** *Step 2: Calculate CENT for all poisoned samples*
>
> ** *Step 3: Sort poisoned samples and recover $n$ samples with the lowest CENT to benign*
>
> ** *Step 4: Obtain poisoned* $\hat{D}^{(t)}$ *with* $(0.5\times|D|-n)$ *poisoned samples*
>
> *Done*
>
> Due to the short time, we focus on BadNet and Blend with $T=5$ and obtain following results for CIFAR10, in which the greedy seems not to yield a strong, stealthy attack.
>
> |Attack|$\rho$|ACC|ASR|
> |-|-|-|-|
> |BadNet|5|94.50|1.03|
> ||1|94.62|1.00|
> |Blend|5|94.85|0.94|
> ||1|94.45|3.50|
>
> ### Response to W4:
>
> For low-redundancy tasks such as Tiny-ImageNet, more data is needed to match the accuracy of clean-dataset training.
> ABCS determines an appropriate coreset size adaptively, ensuring sufficient data coverage.
> While the accuracy drops by ~1%, the reduction in ASR is substantial (down to 0.37% on average).
>
> ### Response to Q2:
>
> Coreset selection is meant to preserve natural accuracy despite data reduction [6,7].
> While multiple selection criteria achieve high ACC, we propose CENT-based measure to additionally exclude poisoned samples.
>
> In Section F, we validate the coreset coverage. ABCS produces a coreset covering all classes (cf. Figure 19) and in particular includes more hard-to-learn ones intrinsically with lower ACC (cf. Figure 6(a))
>
> ---
> [1] Wei et al., "To smooth or not? When label smoothing meets noisy labels," ICML 2022.
>
> [2] Di et al., "Label Smoothing Improves Machine Unlearning," ICLR 2026.
>
> [3] Natarajan et al., "Learning with noisy labels," NeurIPS 2013.
>
> [4] Huang et al., "Uncertainty-aware learning against label noise on imbalanced datasets," AAAI 2022.
>
> [5] Kohler et al., "Uncertainty based detection and relabeling of noisy image labels," CVPR-W 2019.
>
> [6] Coleman et al., "Selection via Proxy: Efficient Data Selection for Deep Learning," ICLR 2020.
>
> [7] Paul et al., "Deep learning on a data diet: Finding important examples early in training," NeurIPS 2021.

---

> > ### Author Rebuttal · Reviewer_faaf · 2026-04-03
> >
> > The authors replied to my questions, but this did not completely dispel my doubts.
> >
> > Regarding theoretical foundations and worst-case handling, the authors reasonably justify defense robustness via an attacker trade-off logic and cleverly use label smoothing to explain sample separation.   However, this mechanistic explanation cannot replace formal mathematical bound proofs.   Moreover, limiting worst-case scenarios strictly to label randomization is overly narrow, failing to cover complex, extreme white-box attacks.
> >
> > For high-intensity adaptive attacks, the authors impressively hand-coded a greedy search algorithm tailored to their CENT metric.   However, the test ran for only five iterations without reaching convergence.   More importantly, they avoided combining this strategy with the challenging WaNet, testing only on weaker attacks like BadNet and Blend, which significantly weakens the robustness evidence against high-intensity white-box attacks.
> >
> > Addressing the ACC decline in complex tasks, the authors reasonably frame it as a necessary security-utility trade-off for a low attack success rate, which is convincing at the current scale.   However, lacking concrete mitigation strategies, this coreset-selection-based defense mechanism could face severe scalability and utility loss risks when deployed in large-scale real-world scenarios requiring high data coverage.

---

> > > ### Author Response · Authors · 2026-04-04
> > >
> > > Thank you for the response to our rebuttal.
> > > Below, we aim to clarify the statements regarding those remaining concerns.
> > >
> > > ### Concern of label randomization
> > >
> > > We acknowledge that empirical evaluation and reasoning cannot replace formal mathematical bound proofs.
> > > However, we would like to emphasize that we do not restrict the worst-case scenario solely to label randomization.
> > > Instead, we use it as a representative strategy to model the setting in which the adversary aims to increase the learning difficulty of the backdoor.
> > >
> > > Moreover, the dichotomy between data separability and backdoor effectiveness, revealed by label randomization, can be interpreted as a relation of learning gradients for the clean task and the backdoor task.
> > > To this end, Zhang et al.[1] measure the orthogonality, that is, the angle between gradient directions.
> > > They formulate backdoor learning as a two-stage continual learning process, in which the backdoor task is learned prior to the clean task.
> > > In the loss landscape, poisoned samples exhibit gradients highly orthogonal to clean samples and Theorem 3.4 shows that *"the learned backdoor behaviors during the first phase will persist and remain unaffected during the second stage"* due to this high orthogonality [1].
> > >
> > > Since well-learned samples have low loss-values and low prediction-uncertainty, the backdoor's fast learning/manifestation ensures consistently low uncertainty for poisoned sample, making them more distinguishable from benign and hard-to-learn samples.
> > > While not a theoretical proof, this provides another theoretic insight, supporting the use of CENT as a robust selection criterion.
> > >
> > > **Worst-Case Attacks**
> > >
> > > Orthogonality determines **how easily the backdoor task can coexist with the clean task**:
> > > - High orthogonality enables fast and robust backdoor learning;
> > > - Low orthogonality leads to slower and less stable backdoor learning.
> > >
> > > A worst-case attack would reduce the orthogonality, thereby increasing the backdoor's learning difficulty.
> > > Label randomization serves exactly this purpose.
> > >
> > > During naive training, we measure the orthogonality under different levels of label randomization using simple attacks, BadNet and Blend, on CIFAR-10.
> > > Compared to the baseline (ratio=0%), stronger label randomization degrades orthogonality more, increasing similarity between backdoor and clean tasks, which affects ABCS's defense.
> > > However, as shown in Table 14, the increased task similarity unavoidably degrades the effectiveness of the backdoor itself.
> > >
> > > |Attack|Randomization Ratio|Epoch-5|Epoch-10|Epoch-15|Epoch-20|Epoch-25|Epoch-30|Epoch-35|Epoch-40|
> > > |-|-|-|-|-|-|-|-|-|-|
> > > |BadNet|0%|47.4|73.4|75.7|78.0|76.7|74.9|78.5|74.3|
> > > ||30%|43.7|71.9|71.7|72.2|70.9|73.8|71.5|71.3|
> > > ||50%|46.6|69.1|70.7|70.6|66.8|70.6|68.8|69.7|
> > > |Blend|0%|54.7|60.0|59.8|61.0|59.7|61.0|62.0|64.4|
> > > ||30%|53.9|57.5|58.8|58.9|61.5|59.0|60.7|61.4|
> > > ||50%|49.4|55.7|56.4|58.0|58.7|59.4|59.1|58.3|
> > >
> > > [1] Zhang et al., "Exploring the Orthogonality and Linearity of Backdoor Attacks," IEEE S&P 2024.
> > >
> > >
> > > ### Insufficient evaluation with greedy-search-based adaptive attack
> > >
> > > Please note that not considering WaNet was down to the limited time for the rebuttal.
> > > We have thus now extended our experiment in this regard and also perform $T=10$ iterations each, which has been shown to converge.
> > > We find that more iterations do not yield more effective backdoors against ABCS.
> > > The adaptive attacks in Section E seem to indeed be more challenging to ABCS.
> > >
> > > |Attack|$T$|$\rho$|ACC|ASR|
> > > |-|-|-|-|-|
> > > |BadNet|5|5|94.50|1.03|
> > > |||1|94.62|1.00|
> > > ||10|5|94.39|0.92|
> > > |||1|94.90|0.88|
> > > |Blend|5|5|94.85|0.94|
> > > |||1|94.45|3.50|
> > > ||10|5|94.43|1.46|
> > > |||1|94.63|4.64|
> > > |WaNet|5|5|94.47|3.16|
> > > |||1|94.03|8.06|
> > > ||10|5|94.55|2.35|
> > > |||1|93.72|8.92|
> > >
> > > ### Security-utility trade-off
> > >
> > > It is difficult to assess large-scale real-world scenarios.
> > > We thus additionally consider a low-redundancy scenario where overly aggressive coreset selection may degrade the utility of training dataset.
> > > For this, we extract a 50% coreset from the original CIFAR-10 training data using CENT and then poison it as the final training dataset.
> > > We then obtain the following results for ABCS:
> > >
> > > |Attack|Defense|ACC|ASR|$\rho_{bng}$|$r_{se}$|
> > > |-|-|-|-|-|-|
> > > |BadNet|No-Defense|94.68|98.44|/|/|
> > > ||ABCS|93.52|1.20|0.00|77.04|
> > > |Blend |No-Defense|94.72|89.84|/|/|
> > > ||ABCS|93.14|1.34|0.02|74.19|
> > > |IAB|No-Defense|94.62|96.87|/|/|
> > > ||ABCS|92.85|1.07|0.02|76.20|
> > > |WaNet|No-Defense|93.92|71.86|/|/|
> > > ||ABCS|92.02|4.40|1.38|75.48|
> > >
> > > ABCS remains effective in selecting backdoor-free coresets, while ensuring sufficient data coverage is more challenging.
> > > Compared to the "No-Defense" baseline, ABCS selects a slightly smaller coreset, causing a small drop in ACC.
> > > Importantly, it automatically converges to a large selection ratio $r_{se}$ (relative to the training coreset), aiming to retain as many samples as possible to preserve accuracy.
> > > This behavior is consistent with Tiny-ImageNet and highlights ABCS's adaptability to low-redundancy scenarios.

---

### Decision · Program_Chairs · 2026-04-30

**Decision:**

Accept (regular)

**Comment:**

This paper introduces a training-time defense mechanism for backdoor attacks on neural networks based on a coreset selection approach called ABCS (Anti-Backdoor Coreset Selection). The proposed method constructs a backdoor-free coreset by tracking the cumulative entropy of each sample during training. The key idea is that poisoned samples are typically learned faster and with lower uncertainty than benign ones, so analyzing these learning dynamics helps identify informative and likely benign training data points. The proposed method further improves separation by iteratively unlearning selected samples and applying label-smoothed training to make benign and poisoned samples better distinguishable. After determining the coreset with this approach, the model retrained on the resulting coreset achieves strong robustness against backdoor attacks without sacrificing clean accuracy. Evaluations across various backdoor attacks show that the method outperforms several prior defenses while remaining efficient and also not requiring benign reference data to facilitate this defense.

The paper went through productive discussions during the rebuttal phase, and the reviewers unanimously acknowledged the value of the paper's contribution. One concern that was raised was the lack of a theoretical framework for the defense mechanism. The authors acknowledged this limitation and provided an empirical justification grounded in prior work on label smoothing, showing that the unlearning mechanism consistently reduces ASR across various settings and that the method is robust in practice. Another concern clarified during the discussions was the unexpectedly low ASR in text classification tasks, which suggested potential evaluation artifacts. The authors clarified this by pointing to the cause to be a class distribution shift introduced by their coreset selection method with an empirical demonstration. The authors have also clarified several other empirical questions, e.g., regarding the lack of evaluations against an optimization-based backdoor attack, and comparisons against existing test-time defenses, by providing additional experiments to address these questions. Authors acknowledged several of these rebuttal clarifications to be added with discussions in their revised manuscript, also including better clarification of the details of how ABCS is implemented, as well as moving some of the important adaptive attack evaluations from the appendix to the main paper.

In general, the AC acknowledges that the rebuttal was effective, with the authors successfully demonstrating the novelty and rigor of their work. Overall, the paper presents an interesting contribution to the field of ML security, and it is very well presented with well-structured experimental evaluations. Thus, the AC recommends acceptance of this paper.